# Ultrafast (400 Hz) network oscillations induced in mouse barrel cortex by optogenetic activation of thalamocortical axons

**Hang Hu, Rachel E Hostetler, Ariel Agmon\***

Department of Neuroscience, West Virginia University School of Medicine, Rockefeller Neuroscience Institute, Morgantown, United States

**Abstract** Oscillations of extracellular voltage, reflecting synchronous, rhythmic activity in large populations of neurons, are a ubiquitous feature in the mammalian brain, and are thought to subserve important, if not fully understood roles in normal and abnormal brain function. Oscillations at different frequency bands are hallmarks of specific brain and behavioral states. At the higher end of the spectrum, 150-200 Hz ripples occur in the hippocampus during slow-wave sleep, and ultrafast (400-600 Hz) oscillations arise in the somatosensory cortices of humans and several other mammalian species in response to peripheral nerve stimulation or punctate sensory stimuli. Here we report that brief optogenetic activation of thalamocortical axons, in brain slices from mouse somatosensory (barrel) cortex, elicited in the thalamorecipient layer local field potential (LFP) oscillations which we dubbed "ripplets". Ripplets originated in the postsynaptic cortical network and consisted of a precisely repeating sequence of 2-5 negative transients, closely resembling hippocampal ripples but, at ~400 Hz, over twice as fast. Fast-spiking (FS) inhibitory interneurons fired highly synchronous 400 Hz spike bursts entrained to the LFP oscillation, while regular-spiking (RS), excitatory neurons typically fired only 1-2 spikes per ripplet, in antiphase to FS spikes, and received synchronous sequences of alternating excitatory and inhibitory inputs. We suggest that ripplets are an intrinsically generated cortical response to a strong, synchronous thalamocortical volley, and could provide increased bandwidth for encoding and transmitting sensory information. Importantly, optogenetically induced ripplets are a uniquely accessible model system for studying synaptic mechanisms of fast and ultrafast cortical and hippocampal oscillations.

**\*For correspondence:**
aric.agmon@gmail.com

**Competing interest:** The authors declare that no competing interests exist.

## Editor's evaluation

This study examines the potential neuronal basis for generating ultrafast oscillations (250-600Hz) in the cortex evoked by optogenetic stimulation of thalamocortical afferents in ex vivo brain slices. The authors proposed that these oscillations correlate with sensory stimulation and may be relevant for the perception of relevant sensory inputs and they combined ex-vivo whole-cell patch-clamp recordings, local field potential (LFP) recordings, and optogenetic activation of thalamic afferents to generate ripple-like extracellular waveforms in the cortex, referred to as "ripplets." The authors described the sequences of RS and FS neuron discharge and how they phase-locked to the ripplet, providing a model for the cellular mechanism generating the ripplet. The authors also cited the literature about ultrafast oscillations and carefully compared the novel ripplets to the well-known hippocampal ripples.

## Introduction

A ubiquitous feature of neuronal networks, in the brains of multiple mammalian species throughout the evolutionary tree, is their propensity to engage in large-scale, synchronous and rhythmic patterns of electrical activity, reflected as oscillations in the local field potential (LFP) (*Buzsáki et al., 2013*). LFP oscillations occur at frequencies spanning four or more orders of magnitude, with oscillations at different frequency bands often nested together and co-modulating each other (*Steriade, 2006*). Different oscillatory patterns are hallmarks of specific brain and behavioral states; for example, theta rhythms (4–9 Hz) are typical of exploratory behavior, sleep spindles (~10 Hz) occur during certain sleep stages, and gamma oscillations (30–90 Hz) are the hallmark of cognitive activity and sensory processing (*Lopes da Silva, 2013*; *Singer, 2018*). Even faster, transient 150–200 Hz 'ripples' in the pyramidal cell layer of the hippocampus occur during quiet immobility and slow-wave sleep (*Bragin et al., 1999a*; *Bragin et al., 1999b*; *Csicsvari et al., 1999a*), and are thought to be critical for memory consolidation (*Girardeau and Lopes-Dos-Santos, 2021*). Extracellular voltage signals, including oscillations, are driven by currents entering (sinks) or leaving (sources) the intracellular brain compartment, and reflect both synaptic activity and action potential firing (*Mitzdorf, 1985*; *Liebe et al., 2011*; *Buzsáki et al., 2012*; *Schomburg et al., 2012*; *Reimann et al., 2013*; *Pesaran et al., 2018*). Indeed, excitatory and inhibitory neurons in the hippocampus fire at precise phases of theta, gamma, and ripple oscillations in a subtype-specific manner (*Klausberger et al., 2003*; *Klausberger and Somogyi, 2008*; *Cardin, 2018*), and synaptic interactions between excitatory and inhibitory cells are implicated in the generation of oscillations across the spectrum, although the details may vary between frequency bands, between different brain areas, and even between cortical layers (*Kopell et al., 2010*; *Wang, 2010*; *Whittington et al., 2018*).

At the high end of the frequency range, ultrafast, 250–600 Hz oscillations have been observed in both hippocampus and neocortex. In the hippocampus, such oscillations (often called 'fast ripples') are almost exclusively paroxysmal (reviewed in *Gulyás and Freund, 2015*; *Lévesque and Avoli, 2019*), although their underlying cellular and network mechanisms remain under debate (*Dzhala and Staley, 2004*; *Foffani et al., 2007*; *Engel et al., 2009*). In the neocortex, however, non-paroxysmal 600 Hz wavelets have been repeatedly observed in scalp recordings from human subjects in response to peripheral nerve stimulation (reviewed in *Curio, 2000*). Similar 600 Hz LFP oscillations in response to peripheral nerve stimulation were also observed in monkeys and piglets (*Ikeda et al., 2002*; *Baker et al., 2003*). Notably, LFP oscillations in the 300–500 Hz range were also consistently observed in rats in response to whisker deflections or to auditory clicks (*Jones and Barth, 1999*; *Jones et al., 2000*; *Jones and Barth, 2002*); however, their underlying mechanisms and functional significance remain largely unexplored.

We recorded ex vivo extracellular and intracellular activity evoked by optogenetic activation of channelrhodopsin (ChR2)-expressing thalamocortical axons in thalamorecipient layers of mouse somatosensory (barrel) cortex. We report here that brief light pulses (as short as 1 ms) elicited a transient (<25 ms), highly reproducible LFP oscillation or 'ripplet,' which closely resembled hippocampal ripples but, at ~400 Hz, was more than twice as fast. Paired whole-cell recordings from fast-spiking (FS) inhibitory interneurons and regular-spiking (RS) excitatory cells revealed precise phase relationships between FS spikes, RS spikes and ripplets, and between FS spikes and sequences of alternating EPSCs and IPSCs in RS cells, suggesting that phasic RS→RS and RS→FS excitation underlies ripplet generation. Ripplets may be a stereotypical cortical response to strong, punctate stimuli that activate thalamocortical afferents with a high degree of synchrony and could be used by the cortex to encode specific features of a sensory event. Importantly, optogenetically evoked ex vivo ripplets are a uniquely accessible model for studying the synaptic basis of fast and ultrafast network oscillations, including hippocampal ripples.

## Results

We recorded extracellular and intracellular responses in layer 4 (L4) of barrel cortex brain slices, evoked by widefield optogenetic activation of thalamocortical synapses. Slices were prepared from brains of 3–7-postnatal-week-old mice of both sexes expressing ChR2 in the somatosensory thalamus (see 'Materials and methods'). We applied brief light pulses through the epi-illumination path of the microscope, likely activating most of the thalamocortical axons and terminals within the surrounding

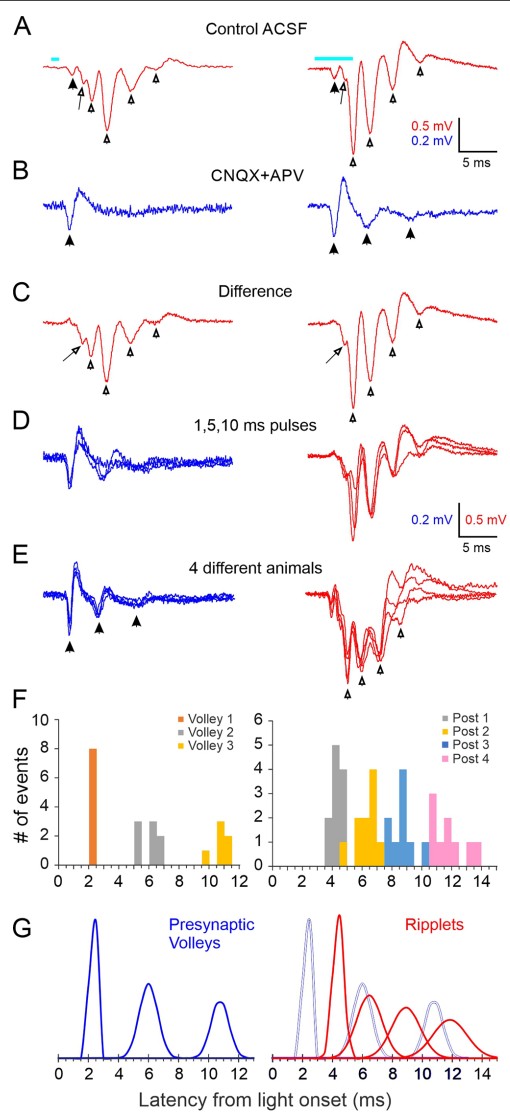

**Figure 1.** Local field potential oscillations (ripplets) induced by optogenetic thalamocortical activation. (**A**) Averaged ripplets evoked in an example slice; light pulses (1 and 5 ms duration for left and right traces, respectively) represented by cyan bars above traces, and apply to (**A**–**C**). Solid arrowhead indicates the early presynaptic TC volley; hollow arrowheads indicate the peaks of the postsynaptic components; the arrow points to the hump on the rising phase of the first postsynaptic transient. (**B**) The iGluR-independent (i.e. presynaptic) component: same stimuli as in (**A**) but in the presence of CNQX + APV. Arrowheads indicate a single presynaptic volley in response to a 1 ms stimulus and three volleys (of decreasing amplitude and coherence) with a 5 ms stimulus. (**C**) The difference between (**A**) and (**B**), revealing the purely postsynaptic component of the ripplet. Symbols as in (**A**). (**D**) Superposition of the presynaptic (blue, left) and postsynaptic (red, right) components of the ripplets at three stimulus durations. (**E**) Superposition

*Figure 1 continued on next page*

L4 barrel. To verify that ChR2 in L4 was expressed by thalamocortical and not intracortical axons, we imaged and quantified the extent of tdTomato reporter expression in the cortex (*Figure 1—figure supplement 1*). Within the barrel cortex, there was a sparse population of reporter-expressing cell bodies in L2/3 and L5, but virtually none within L4 (*Figure 1—source data 1*). Since the thalamus and L4 itself are the two main sources of excitatory input to L4 neurons (*Feldmeyer et al., 1999*; *Petersen and Sakmann, 2000*; *Lefort et al., 2009*), we conclude that the network activity we observed was predominantly evoked by thalamocortical activation, although we cannot rule out that a small number of corticothalamic neurons in L6, which send axonal collaterals to L4 (*White and Keller, 1987*; *Kumar and Ohana, 2008*), also expressed ChR2 and provided a minor excitatory contribution.

## Optogenetic activation of thalamocortical axons evoked ripple-like extracellular wavelets

A brief light pulse (1–10 ms, typically 2–5 ms) evoked a stereotypical field potential wavelet in L4 (*Figure 1A*). In this and all other electrophysiological traces, the light-evoked response is represented by a 25-ms-long record beginning 1 ms before light onset, and the timing of the light pulse is indicated by a cyan bar above or below the trace. At all stimulus durations, the LFP waveform consisted of an initial low-amplitude negative component (solid arrowhead), followed by 2–5 (typically 3–4) larger negative transients (hollow arrowheads) riding on a slow, positive-going envelope. In the example slice in *Figure 1A*, the fourth transient was barely discernible in response to a 1 ms light pulse (left), but became noticeably larger with a 5 ms pulse (right). A low-amplitude negative shoulder or 'hump' (arrow) was often observed on the rising phase of the first large transient. These optogenetically evoked waveforms were reminiscent of hippocampus ripples, LFP wavelets recorded from the cell body layer of areas CA1 or CA3; we therefore named them RIPPle-Like Extracellular Transients or 'ripplets'.

Synchronous spike volleys in thalamocortical terminals generate relatively large current sinks in layer 4, and these are discernible as negative transients in the LFP (*Morin and Steriade, 1981*; *Agmon and Connors, 1991*; *Swadlow et al., 2002*; *Bruno et al., 2003*). To reveal any presynaptic thalamocortical components in the ripplets, we blocked fast ionotropic glutamate receptors (iGluRs) pharmacologically with CNQX + APV.

*Figure 1 continued*

of presynaptic components (blue, left) and ripplets (red, right) from slices from four different animals, normalized to the same maximal amplitudes to facilitate comparison of their time course. (**F**) Histogram of peak times (from light onset) of presynaptic volleys (left) and of ripplet transients (right) in slices from 8 and 11 animals, respectively (5 animals are included in both plots). Presynaptic volleys peaked at 2.3 ± 0.07, 6.0 ± 0.22, and 10.8 ± 0.24 ms from light onset. Ripplet transients peaked at 4.4 ± 0.11, 6.5 ± 0.26, 8.9 ± 0.30, and 11.9 ± 0.38 ms. (**G**) The histograms from (**F**) modeled as normal distributions, with the same mean, SD and integral as the corresponding component. Left, presynaptic volleys; right, ripplet transients (red) with presynaptic volleys overlaid as gray outlines for comparison.

The online version of this article includes the following source data and figure supplement(s) for figure 1:

**Source data 1.** Counts of Cre-expressing neurons in the KN282 mouse.

**Figure supplement 1.** Cre-expressing neurons in the KN282 mouse.

**Figure supplement 2.** TTX blocks the extracellular thalamocortical volleys.

This eliminated the large negative transients but left unchanged the early negative component (*Figure 1B*, left panel, solid arrowhead), which therefore reflected the initial thalamocortical volley. Interestingly, the presynaptic response to 5 ms or longer light pulses revealed one or two additional presynaptic transients of decreasing amplitudes (*Figure 1B*, right panel, solid arrowheads), suggesting that thalamocortical axons fired additional volleys of decreasing magnitude and coherence. Subtracting the waveforms in CNQX + APV from those in control ACSF isolated the postsynaptic component of the response (*Figure 1C*). Superposition of the isolated presynaptic (*Figure 1D*, left) and postsynaptic (*Figure 1D*, right) contributions at 1, 5, and 10 ms stimulus durations demonstrated that prolonging the light pulse added late components to the response waveform but did not affect the timing of the earlier peaks. To confirm that the iGluR-independent signals reflected action potentials, we added TTX to the bath, which blocked all remaining responses except for a small square waveform (*Figure 1—figure supplement 2*). The latter was coincident with the light pulse and was most likely an electrical or optoelectrical artifact, and is subtracted from all traces in *Figure 1*.

Ripplet waveforms, as well as their isolated presynaptic components, were remarkably consistent between animals in their temporal structure, as illustrated in *Figure 1E* by the superposition of responses in slices from four different animals, drawn at different vertical scales to facilitate time course comparison. To quantify the variability in the temporal structure of ripplets between animals, we measured the peak times (relative to light onset) of the isolated presynaptic volleys (*Figure 1F*, left; eight slices from eight animals tested in CNQX + APV) and of the postsynaptic transients (*Figure 1F*, right; 11 slices from 11 animals with ripplet amplitudes ≥0.5 mV). Standard deviations (SDs) of the presynaptic volleys ranged from 0.2 to 0.6 ms, and SDs of ripplet peaks ranged from 0.3 to 1.0 ms. Calculated over the three volleys, the presynaptic axons appeared to fire at an average frequency ( = 1/average ISI) of 239 ± 6 Hz. This is consistent with the observed burst frequency in thalamocortical neurons of the same mouse genotype when activated optogenetically ex vivo, which rarely exceeds 300 Hz (*Hu and Agmon, 2016*, their Figure 4E). Calculated over the four ripplet peaks, ripplet frequency averaged 408 ± 15 Hz, that is, nearly twice as fast as the apparent presynaptic volleys. To facilitate comparison of presynaptic components and ripplets, we modeled the histogram peaks in *Figure 1F* as Gaussians with the same mean, SD, and integral as that of the respective event (*Figure 1G*). Overlaying the Gaussians (*Figure 1G*, right) highlighted the lack of temporal coherence between the presynaptic volleys and ripplets, indicating that ripplets were not a direct, 1:1 response to bursts in thalamocortical axons but were likely generate de novo within the L4 network.

## Optogenetic stimulation of thalamocortical axons evoked precisely timed FS spike bursts

Since ripplets required intact glutamatergic neurotransmission, they most likely reflected population activity within L4; but in which cells? To answer this question, we examined light-induced whole-cell responses in inhibitory FS interneurons and excitatory, RS cells, two subclasses known to receive direct thalamocortical inputs (*Agmon and Connors, 1992*; *Gibson et al., 1999*; *Porter et al., 2001*; *Bruno and Simons, 2002*; *Gabernet et al., 2005*; *Sun et al., 2006*; *Cruikshank et al., 2007*; *Shigematsu et al., 2019*). These two subtypes are readily identifiable in slice recordings by their firing patterns

and also fall into distinct, nonoverlapping clusters in post hoc analysis of their electrophysiological parameters (*Figure 2—figure supplement 1*, *Figure 2—source data 1*).

A brief (typically 2–5 ms) light pulse at 60–90% maximal intensity elicited a strong (10–30 mV) depolarization in L4 FS cells, which, in 80% of cells (N = 44 cells from 35 animals), gave rise to a stereotypical burst of 2–7 (typically 3–5) spikes. Representative bursts in an example FS cell, evoked by 2 and 5 ms light pulses, are illustrated in *Figure 2A and B*. Spiking almost never persisted beyond the illustrated 25 ms time window, although the underlying subthreshold depolarization often lasted for 50 ms or more. We performed detailed analysis on all FS cells that consistently fired bursts of four or more spikes (25 cells from 20 animals). Bursts in this dataset had three distinguishing properties:

1. Very high frequency: Averaged over these 25 cells, the first three interspike intervals (ISIs) were 2.4 ± 0.05, 2.2 ± 0.08, and 2.7 ± 0.13 ms (mean ± SEM), respectively, for an average burst frequency ( = 1/average ISI) of 418 ± 9.5 Hz, not significantly different from the extracellular ripplet frequency (p=0.56).
2. Very high temporal precision: Bursts in a given cell were precisely reproducible, as illustrated in *Figure 2A and B* by the near-perfect registration of consecutive sweeps, the alignment of spikes in the raster plots and the very low jitter in spike times indicated above the raster plot. In the full dataset, spike jitter averaged 23 ± 4, 63 ± 10, 110 ± 12, and 183 ± 21 μs for the four spikes, which translates to coefficients of variation (CV = SD/mean) of 0.7 ± 0.1%, 1.1 ± 0.2%, 1.4 ± 0.1% and 1.8 ± 0.2%, respectively. This low jitter is also illustrated in *Figure 2H and I*, in which the vertical extent of the boxed regions indicates 10–90th percentiles of spike time jitter and CV, respectively.
3. High animal-to-animal reproducibility: Across slices from different animals, the distributions of peak spike times for each of the four spikes in the burst was very narrow: 3.1 ± 0.06, 5.5 ± 0.08, 7.7 ± 0.12, and 10.3 ± 0.14 ms after light onset. This is also illustrated in *Figure 2H and I*, in which the horizontal extent of the boxed regions indicates 10–90th percentile of spike times.

Since thalamocortical axons also form a lower tier of terminations in L5B (*Frost and Caviness, 1980*; *Herkenham, 1980*; *Agmon et al., 1993*; *Meyer et al., 2010*), we also tested for occurrence of light-evoked spike bursts in infragranular FS cells. Brief light pulses evoked in L5B FS interneurons spike bursts that were indistinguishable, in spike number and frequency, from those in L4; of 13 cells stimulated with 2 ms light pulses, 11 cells from nine animals fired consistently four spikes/stimulus, at an average frequency of 405 ± 13 Hz (*Figure 2—figure supplement 2*). In the remainder of this study, we focus on L4.

To examine if burst patterns were dependent on stimulus parameters (light duration and intensity), we first compared bursts elicited by 2 and 5 ms light pulses. In the example cell in *Figure 2A*, a 2 ms stimulus elicited a precisely reproducible four-spike burst (10 superimposed red traces); increasing stimulus duration to 5 ms left the timing and jitter on these four spikes nearly unchanged but occasionally elicited a fifth, long-latency spike (arrowhead) with considerably higher jitter compared to the first four spikes (blue traces; note increased jitter in the raster plot in the lower panel). We refer to such spikes, with higher jitter and occasional failures, as 'low-reliability spikes.' All seven FS cells tested at both stimulus durations fired more spikes in response to a 5 ms compared with 2 ms stimulus, with a median increase of one spike/stimulus (*Figure 2C*) but with no appreciable change in timing and precision of the earlier spikes, as seen by the identical median values of the second spike jitter and the first ISI at the two stimulus durations (*Figure 2E*).

We then compared the number and precision of spikes in response to different light intensities. Bursts elicited by 20 and 90% intensity, in the same example cell from *Figure 2A*, are illustrated in *Figure 2B*. At the lower intensity, there was again one less spike, but in addition the jitter of all spikes except the first was >3× higher. In a subsample of 14 cells tested with 5 ms light pulses at five different intensities (10, 20, 40, 60, and 90% of maximal intensity), the average number of spikes/trial increased from 2.9 at 10% to 4.8 at 90% intensity (*Figure 2D*). As seen by comparing the second spike jitter and first ISI between at 20 and 90% light pulses (*Figure 2F*), overall spike timing did not differ between the two intensities (p=0.38) but at lower intensities spikes were much less precise (second spike jitter: 106 ± 13 vs. 43 ± 6 μs, p<0.0001). That ISIs were largely independent of stimulation intensity suggested that the burst pattern was not intrinsically generated in the FS cell but was imposed on it by extrinsic inputs. Consistent with this conclusion, reducing the light level resulted in some cells in spikes dropping out sporadically, revealing apparent subthreshold EPSPs that remained aligned with the temporal structure of the burst (*Figure 2G*, guidelines). That EPSPs elicit FS spikes with such

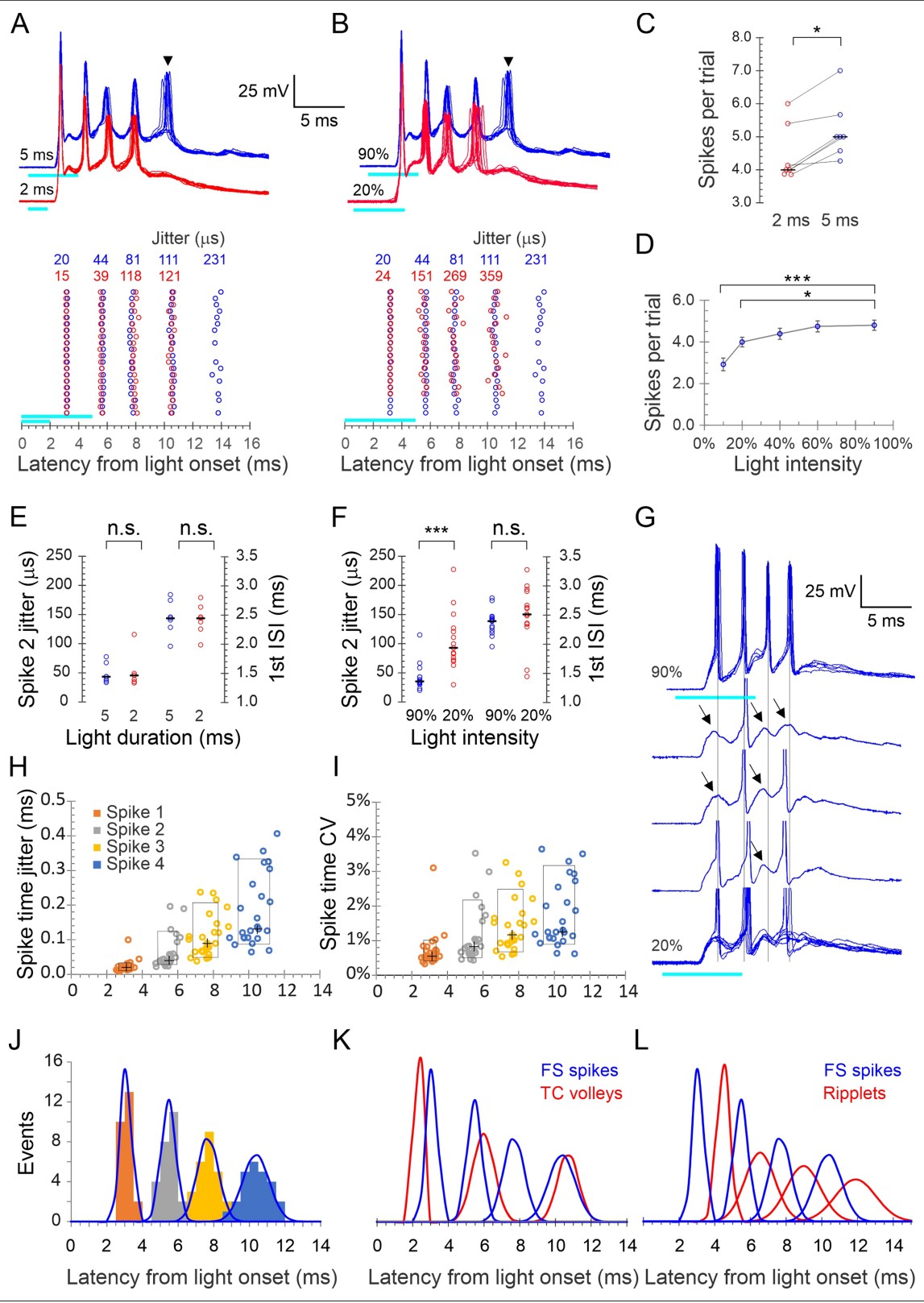

**Figure 2.** Precise light-induced spike bursts in fast-spiking (FS) interneurons. (**A**) Upper panel: spike bursts evoked in an example L4 FS interneuron by 5 ms (blue traces) and 2 ms (red traces) light pulses (cyan bars) at 90% intensity; 10 consecutive sweeps repeated at 8 s intervals at each duration. Arrowhead indicates a late, low-reliability spike evoked by the longer light pulse. Lower panel: raster plots of 20 consecutive bursts from the same neuron, evoked by 2 and 5 ms pulses (red and blue symbols, respectively). Spike time jitters (SD of spike peak times), in μs, are indicated above the

*Figure 2 continued on next page*

*Figure 2 continued*

raster, in the respective colors. (**B**) As in (**A**) but for bursts evoked by 5ms light pulses at 20 and 90% intensities (red and blue traces and symbols, respectively). (**C**) Average number of spikes/stimulus in seven FS interneurons tested with both 2 and 5 ms light pulses. Medians indicated by horizontal black lines. *p=0.016, sign test. (**D**) The number of spikes/stimulus fired by 14 FS interneurons tested at 4–5 different light intensities. Error bars indicate SEM. ***p<0.0001; *p=0.02. (**E**) Jitter of the second spike in the burst, and the first interspike interval (ISI), compared between 5 and 2 ms light pulses, from the dataset of panel (**C**); n.s., not significant. (**F**) Same as (**E**), but comparing the lowest and highest light intensities from the dataset of panel (**D**). Note significant difference in spike jitter (***p<0.0001) but not ISI. (**G**) Another example cell stimulated at 90% intensity (upper trace, five superimposed sweeps) and at 20% intensity (lower traces, three single traces and eight superimposed sweeps). Note that at 20% intensity, spikes drop out sporadically and reveal subthreshold excitatory postsynaptic currents (EPSPs) (arrows), with little change in the temporal structure of the burst (vertical guidelines are aligned with the spikes at 90% intensity). (**H**) Spike times in all 25 FS cells with each spike order in a different color, plotted by its mean time measured from light onset (X axis) and by its jitter (Y axis); crosses indicate medians, boxes indicate 10–90th percentile range. (**I**) As in (**H**) but plotted along the Y axis by CV (jitter/mean spike time). (**J**) Histogram of all average spike times using the same color scheme as in (**H**); each peak in the histogram is overlaid by a Gaussian with the same mean, SD, and integral (blue curves). (**K**) The Gaussians from panel (**J**), representing FS spike bursts, superimposed with the Gaussian from *Figure 1G* representing thalamocortical spike volleys (blue and red curves, respectively). (**L**) The Gaussians representing FS spike bursts (blue) and those from *Figure 1G* representing ripplets (red) are superimposed; note the antiphase relationship between FS spikes and ripplets.

The online version of this article includes the following source data and figure supplement(s) for figure 2:

**Source data 1.** Electrophysiological parameters of fast-spiking (FS) and regular-spiking (RS) neurons.

**Figure supplement 1.** Comparison of electrophysiological parameters of fast-spiking (FS) and regular-spiking (RS) neurons.

**Figure supplement 2.** 400 Hz spike bursts in infragranular fast-spiking (FS) interneurons.

high precision is consistent with the properties of excitatory synaptic inputs onto FS interneurons in neocortex and hippocampus (*Geiger et al., 1997*; *Galarreta and Hestrin, 2001*).

What were the temporal relationships between FS spike bursts, thalamocortical volleys, and ripplets? We plotted all FS spike times in our dataset as a histogram and then modeled each spike order as a Gaussian (*Figure 2J*). Overlaying Gaussians representing FS spikes with those representing thalamocortical volleys from *Figure 1G* (*Figure 2K*) revealed a close temporal relationship between the first FS spike and the initial thalamocortical volley, with the postsynaptic spike following the presynaptic volley at a latency of 0.8 ms, consistent with a monosynaptic response. Later FS spikes, however, actually preceded the closest presynaptic volley, consistent with the conclusion above that the additional thalamocortical volleys did not drive the postsynaptic bursts in a 1:1 manner. In contrast, overlaying the distributions of FS spikes and LFP transients (*Figure 2L*) revealed a 1:1 relationship between these events, however, with the two oscillations almost exactly out of phase: each ripplet transient lagged behind the corresponding FS spike peak by 0.44–0.54 of a cycle, translating to 1.0–1.3 ms.

## FS cells synchronized their firing with near-zero lag and submillisecond precision

Given the precise temporal relationship between FS spikes and the population activity reflected as ripplets, one would expect different FS cells in the same barrel to fire in tight synchrony with each other. We therefore examined intra-barrel FS-FS temporal relationships by simultaneous paired recordings. Indeed, when stimulated by a light pulse, simultaneously recorded FS-FS pairs exhibited highly precise spike synchrony. In the two example pairs shown in *Figure 3A and B*, the peaks of the first three spikes in each burst aligned to within 0.1–0.3 ms between the two cells. To quantify the magnitude and precision of pairwise synchrony, we used the Jitter-Based Synchrony Index (JBSI), which is a normalized measure of spike-spike synchrony in excess of that expected by chance (*Agmon, 2012*; see 'Materials and methods'). Synchrony is defined as the fraction of spikes co-occurring within a predetermined 'synchrony window' (SW), and chance synchrony is the average synchrony remaining after shifting each spike in one train by a random jitter ≤J, J = 2·SW. When calculated for decreasing SW values, the JBSI typically increases to a maximum and then precipitously drops off to zero, as J falls below the intrinsic precision of the system and the applied jitter no longer disrupts the pairwise synchrony. We defined 'pairwise precision' as the smallest SW value with JBSI > 0.5. Since neurons can maintain a precise temporal relationship even when they fire at non-zero time lags relative to each other, one can extend this analysis by calculating the JBSI for a range of virtual lags (shifts) of one spike train relative to the other, analogous to a classical cross-correlation histogram; we defined 'pairwise lag' as the virtual lag for which the JBSI was maximized. The full analysis can be represented as a two-dimensional matrix in which the JBSI is plotted as a function of the two parameters, SW and virtual

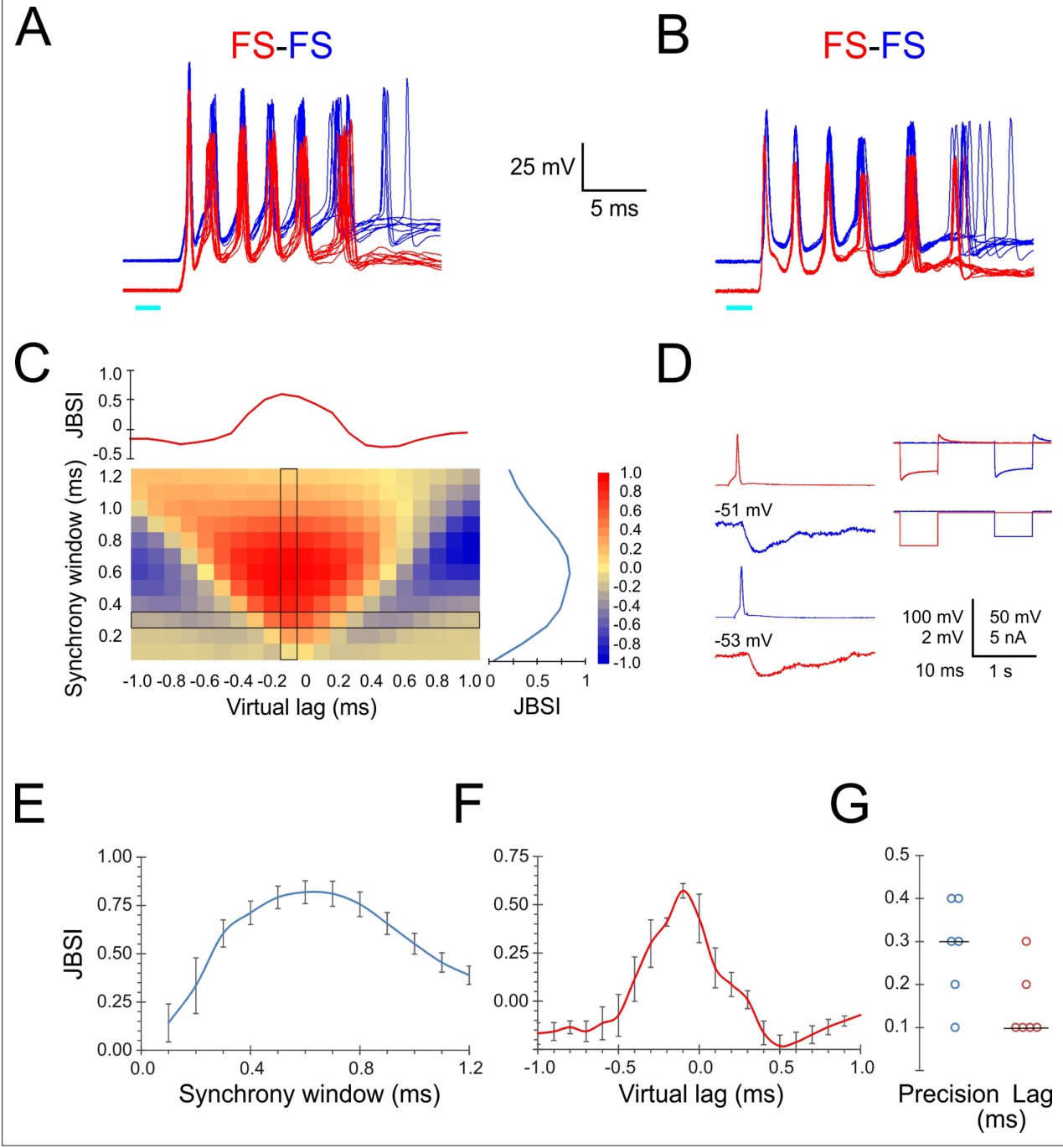

**Figure 3.** Precise firing synchrony between L4 fast-spiking (FS) cells. (**A, B**) Simultaneously recorded spike bursts (eight superimposed sweeps each) from two example FS-FS pairs. Traces from the two cells are displaced vertically for clarity. (**C**) Heat map-coded matrix of Jitter-Based Synchrony Index (JBSI) values calculated from ~300 spikes from each FS cell in (**A**), for a range of synchrony windows (vertical axis) and virtual lags (horizontal axis) in 0.1 ms increments. The plots to the right and above the matrix are cross-sections along the column and row, respectively, indicated by a black border. The matrix element at the intersection corresponds to the smallest synchrony window with JBSI > 0.5 and the highest JBSI value along that row and was used to define the pairwise precision and lag of this pair. (**D**) Averaged traces of reciprocal unitary inhibitory postsynaptic potentials (IPSPs) between the two FS cells in (**B**, left), with holding potentials indicated, and a test for electrical coupling (right); coupling coefficient was 0.8%. (**E**) Average ± SEM of vertical cross-sections (as illustrated in **C**) through the JBSI matrices of 6 FS-FS pairs. (**F**) Same for horizontal cross sections. (**G**) Summary of pairwise precision and lag for the six FS-FS pairs; horizontal lines indicate medians.

lag, as illustrated in *Figure 3C* for the FS pair shown in panel A. Vertical and horizontal cross-sections through the JBSI matrix are plotted to the right and top of the matrix, illustrating how the JBSI varied with SW and virtual lag values, respectively.

We calculated the JBSI matrices for six FS-FS pairs from which at least 100 spikes were recorded per cell. The averaged horizontal and vertical cross-sections through the six matrices are plotted in *Figure 3E and F*, respectively, and the pairwise precision and lag (in absolute values) of the six pairs are plotted in *Figure 3G*. The median pairwise precision and lag were 0.3 and 0.1 ms, respectively. While there is some degree of arbitrariness in the exact definition of pairwise precision, this analysis indicates that FS spikes within a given barrel synchronized with near-zero lag and with sub-millisecond precision.

What accounted for this highly precise FS-FS synchrony? Precise spike synchrony is most often attributed to electrical coupling via gap junctions (*Galarreta and Hestrin, 1999*; *Mancilla et al., 2007*), inhibitory synaptic connectivity (*Gibson et al., 2005*; *Hu et al., 2011*), or common excitatory inputs (*Perkel et al., 1967*; *Sears and Stagg, 1976*; *Alonso et al., 1996*). Unlike rat cortex, electrical coupling between FS interneurons in the mouse is weak, with average coupling coefficient of ~1.5% (*Galarreta and Hestrin, 2002*; *Meyer et al., 2002*; *Hu et al., 2011*; *Hu and Agmon, 2015*). The right panel in *Figure 3D* illustrates how we tested for electrical coupling, which in all six pairs was absent or barely detectable (coupling coefficient < 1%). In addition, inhibitory synaptic connections were observed in only two of the six FS-FS pairs, including the pair of *Figure 3B* (*Figure 3D*, left panels) but not of *Figure 3A*. We conclude by elimination that the precise FS-FS synchrony we observed was most likely driven by common excitatory inputs.

## FS and RS spikes were phase-locked to the ripplet oscillation

To examine directly and in more detail the temporal relationship between FS spike bursts and ripplets, we recorded light-evoked bursts in nine FS cells (from nine animals), simultaneously with the LFP in the same barrel. Three examples from different animals are illustrated in *Figure 4A*, with averaged LFP ripplets (magenta traces) and 10 superimposed intracellular FS bursts (red traces). As indicated by the vertical guidelines, FS spike peaks were closely aligned with the troughs (the positive maxima) in the extracellular waveforms, almost exactly out of phase with the ripplet peaks (the negative maxima), consistent with the conclusion above from the comparison of the population distributions in *Figure 2L*. *Figure 4A* also illustrates that the number of FS spikes in the different slices (3, 4, and 5) was exactly one more than the number of ripplet peaks (2, 3, and 4, respectively), reflecting the fact that an FS spike occurred on every trough, from the one preceding the first LFP peak to the one following the last.

Population activity reflected in the LFP is likely to be dominated by excitatory (RS) cells, which are the majority (~85%) cell type in L4 (*Lin et al., 1985*; *Gabbott and Somogyi, 1986*; *Beaulieu, 1993*). We recorded from RS cells in whole-cell, current-clamp, and voltage-clamp modes to reveal the timing of spikes and synaptic events, respectively, in relation to ripplets and FS bursts. In current-clamp mode, RS cells (*Figure 4B*, blue traces) responded to brief light pulses with a large depolarization (up to 15–20 mV above resting potential), crested by a series of 3–5 apparent EPSPs (slanted arrowheads) and typically giving rise to 1–2 spikes. Of 55 RS cells recorded in current-clamp mode and stimulated at a near-maximal light level, 73% fired at least one, but never more than two spikes/stimulus; the rest either remained subthreshold (11%) or occasionally fired three or more spikes (16%). Spikes evoked by near-maximal stimulus levels were mostly highly precise (*Figure 4B*, left), but spikes arising from late EPSPs were often of lower reliability and occasionally failed (*Figure 4B*, right, vertical arrowhead). Spikes evoked by low light levels would often switch sporadically between EPSPs, a pattern we refer to as 'EPSP hopping' (*Figure 4B*, left, inset). To examine phase relationships between RS spikes and ripplets, we recorded in current-clamp mode from 10 RS cells (from 10 animals) simultaneously with the LFP. As indicated by the vertical guidelines in the two example records in *Figure 4B*, peaks of spikes and of apparent EPSPs in RS cells occurred close to and usually slightly before the LFP negative peaks, that is, in-phase with ripplets. This is reminiscent of hippocampus CA1 pyramidal cells, which fire close to the negative ripple peaks (*Buzsáki et al., 1992*; *Csicsvari et al., 1999b*; *Klausberger et al., 2004*; *Hulse et al., 2016*).

We quantified the phase relationships between ripplets and FS and RS spikes in these two datasets (of 9 and 10 cells from 9 and 10 animals, respectively) by assigning each ripplet trough a phase angle

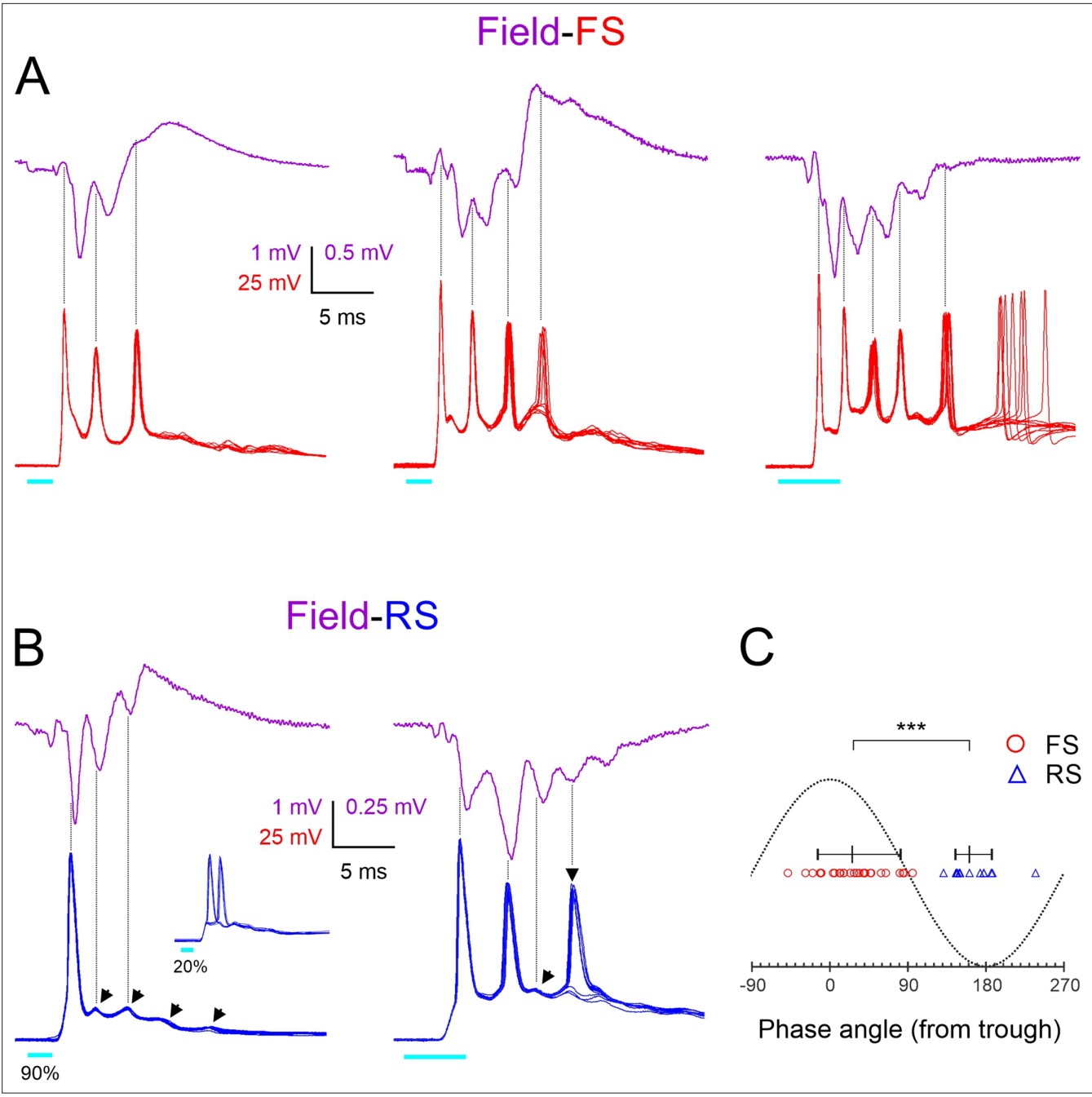

**Figure 4.** Phase-locking of fast-spiking (FS) and regular-spiking (RS) spikes to ripplets. (**A**) Averaged ripplets (upper traces, magenta) and simultaneously recorded FS spike bursts (lower traces, superimposed consecutive sweeps, red) in response to 2 or 5 ms light pulses (cyan bars) in slices from three different animals. Vertical guidelines are aligned with FS spike peaks; note that FS spikes aligned with ripplet troughs. The leftmost two slices were not tested in TTX, and therefore the square-wave stimulus artifact is not subtracted and partially occludes the presynaptic volley. (**B**) As (**A**) but lower traces are from two RS cells. Slanted arrowheads in left panel point to subthreshold excitatory postsynaptic currents (EPSPs). Note that both RS spikes and EPSPs were aligned with ripplet peaks. Inset in left panel illustrates the same cell stimulated at a weaker light intensity, with the spike 'hopping' between the first and second EPSPs. Vertical arrowhead in right panel points to a late, low-reliability spike. (**C**) A phase plot with ripplet troughs and peaks designated as $0^0$ and $\pm180^0$, respectively. FS and RS spike peaks are indicated by red circles and blue triangles, respectively. Horizontal lines with tick marks indicate medians and 10–90th percentile ranges, respectively. ***p<0.0001.

of 0°, and assigning phase angles of –180° and 180° to the negative ripplet peaks preceding and following the trough, respectively. We then assigned a phase angle to each of the 28 FS spikes and 13 RS spikes observed in these experiments by interpolating its time of peak between the nearest ripplet trough and peak (*Figure 4C*). The phase angles of the two cell types were nonoverlapping, with FS spikes occurring at a median phase angle of 26° (i.e. slightly past the trough; 10–90th percentile range: (–14)–82°, translating to a range of 0.65 ms), and RS spikes occurring at a median phase angle of 161° (i.e. slightly before the peak; 10–90th percentile range: 145–187°, translating to a range of ~0.3 ms). Thus, on average, FS spikes preceded RS spikes by ~135° or ~0.9 ms. While this appears inconsistent with the order of firing during hippocampal ripples, in which FS cells lag behind pyramidal cells by about 100° (*Hájos et al., 2013*), 100° of a 180 Hz ripple oscillation amounts to 1.5 ms. Given the 2.4 ms ripplet period, the 0.9 ms delay between L4 FS and RS spikes means that FS spikes (other than the first) occurred 1.5 ms after RS spikes, exactly as in the hippocampus. Notably, a 1.5 ms lag is precisely the time it takes for RS cells to generate monosynaptic EPSPs in FS cells, and for postsynaptic FS cells to reach threshold and fire (*Galarreta and Hestrin, 2001*). Thus, this analysis is consistent with a model by which FS spikes (other than the first, which was clearly driven by thalamocortical excitation) were directly driven by excitation from neighboring RS cells.

## FS spikes and ripplet troughs were phase-locked to EPSC-IPSC alternations in RS cells

To examine the temporal relationships between FS and RS cell responses directly, we performed simultaneous recordings from six FS-RS pairs (from four animals), switching the RS recording mode between current and voltage clamp to reveal both spikes and postsynaptic currents, respectively. When in voltage clamp, we held the RS cells between –50 and –55 mV to distinguish between EPSCs (negative deflections) and IPSCs (positive deflections). We also tested recorded pairs for direct synaptic connections between them. Two example pairs are shown in *Figure 5A and B*, with 10 superimposed traces of FS spike bursts (red) and simultaneously recorded RS postsynaptic currents (blue), and with sequentially recorded RS spikes in the upper trace. Note that the pair in panel *Figure 5A* is illustrated at two different stimulus intensities. The voltage-clamped RS responses revealed a sequence of regularly alternating EPSCs and IPSCs (E's and I's, onsets marked by filled and hollow arrowheads, respectively). These E-I sequences were tightly synchronized between neighboring RS cells in a given barrel (*Figure 5—figure supplement 1*). The dashed vertical guidelines in *Figure 5A and B* are aligned with FS spikes peaks and demonstrate that each FS spike occurred immediately following the onset of an EPSC and before the onset of an IPSC in the RS cell. Although both pairs were synaptically connected in the FS→RS direction (insets), IPSCs in the RS cell were still observed when FS spikes dropped out at the lower stimulus level (rightmost guideline in panel *Figure 5A*, lower traces), indicating that they were evoked by multiple FS cells in addition to the recorded one. Comparing the total charge transfer (the area under the curve) between the voltage-clamp records of RS cells A and B reveals a net inhibitory response in A (positive total charge) and a net excitatory response in B (negative total charge); this large variability in excitatory-to-inhibitory ratio between RS cells was representative of our sample (see also *Figure 5—figure supplement 1*), and is also observed in hippocampal pyramidal neurons during ex vivo ripples (*Ellender et al., 2010*; *Hájos et al., 2013*; *Gan et al., 2017*).

To quantify the temporal relationships between FS spikes and E-I alternations in RS cells, we assigned a phase angle to each of the 16 FS spikes observed in this dataset by defining RS EPSC onsets as 0° and RS IPSC onsets as 180° (*Figure 5C*). The median FS spike phase angle was 59° (10–90th percentile range: 5–106°). Therefore, given the 2.4 ms oscillation period, each FS spike occurred about 0.4 ms after the onset of an EPSC in the RS cell and 0.8 ms before the onset of an IPSC in the same RS cell. The most parsimonious interpretation for these temporal relationships is that RS and FS cells received a barrage of shared EPSPs at the ripplet frequency; these EPSPs elicited short-latency (0.4 ms) spikes in the FS cells, and FS spike volleys, in turn, elicited IPSPs in RS cells after a 0.8 ms monosynaptic delay. As described above, RS spikes occurred, on average, 0.9 ms after the previous FS spike volley, implying that RS cells reached firing threshold nearly simultaneously with the arrival of a wavefront of inhibition, leaving a very narrow 'window of opportunity' for RS cells to fire. The earliest cohort of RS cells to reach spike threshold fired while 'laggards,' cells slightly slower to reach threshold, were preempted from firing by the incoming IPSP volley. This explains why only a fraction

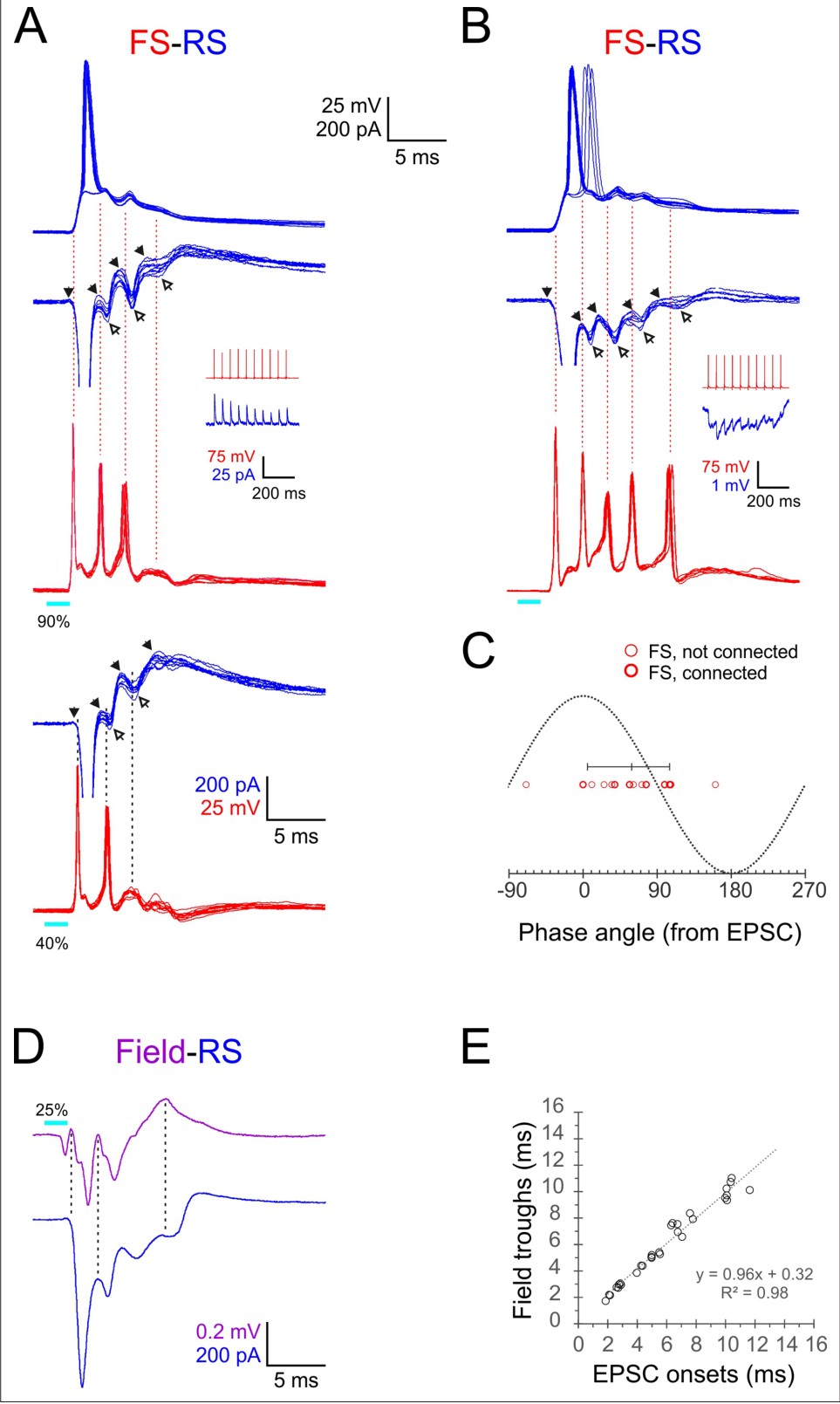

**Figure 5.** Phase-locking of E-I alternations in regular-spiking (RS) cells to fast-spiking (FS) spikes and ripplets. (**A, B**) Superimposed FS spike bursts (red), simultaneously recorded voltage-clamped excitatory postsynaptic currents–inhibitory postsynaptic currents (EPSC-IPSC) alternations (middle traces, blue) and sequentially recorded current-clamp responses from paired RS cells (upper traces, blue), evoked by 2 ms light pulses (cyan bars under

*Figure 5 continued on next page*

*Figure 5 continued*

red traces) in two example FS-RS pairs. Note that the spike in the right RS cell 'hops' between the first and second EPSPs. Solid and empty arrowheads point to EPSC and IPSC onsets, respectively. The first IPSC in both RS cells is occluded by an unclamped spike. Vertical guidelines are aligned with FS spike peaks and illustrate that each FS spike immediately followed an EPSC and preceded an IPSC in the RS cell. Pair in (**A**) was also stimulated at a lower intensity (lower pair of traces), showing that IPSCs persisted even when FS spikes dropped out. Insets illustrate unitary IPSCs (**A**) and IPSPs (**B**) evoked in each RS cell by a 20 Hz train of spikes in the paired FS cell. (**C**) A phase plot of FS spikes relative to the EPSC-IPSC alternations in RS cells, with EPSC and IPSC onsets mapped to $0^0$ and $180^0$, respectively. Phase angles of FS spikes (16 spikes from six pairs) are indicated by red circles, with heavier symbols corresponding to pairs with a direct FS→RS connection; vertical bar indicates median and 10–90th percentile range. (**D**) Averaged voltage-clamped response (blue) of an example RS cell recorded simultaneously with the averaged ripplet in the same barrel (magenta). Vertical guidelines illustrate the correspondence between ripplet troughs and EPSC onsets in the RS cell. (**E**) EPSC onset times plotted against ripplet trough time for eight RS cells in eight slices; equation and $R^2$ value of regression line indicated.

The online version of this article includes the following figure supplement(s) for figure 5:

**Figure supplement 1.** Synchronous E-I sequences in neighboring regular-spiking (RS) cells.

**Figure supplement 2.** Ripplets convert to paroxysmal discharges under GABA_A receptor block.

of RS cells fired on any given cycle and also explains how synchronous RS firing was maintained from one cycle to the next.

To further test the role of inhibition in ripplet oscillations we blocked fast inhibition locally, with the GABA_A antagonist gabazine applied from an extracellular patch pipette placed in the same barrel. As observed in some studies of hippocampus ripples (*Behrens et al., 2007*; *Ellender et al., 2010*), this manipulation converted ripplet-related spiking to prolonged (up to 500 ms) paroxysmal discharges, in both FS and RS cells (*Figure 5—figure supplement 2*).

Lastly, to shed light on the current sinks and sources underlying the extracellular LFP oscillation, we examined directly the temporal relationships between ripplets and E-I sequences in RS cells by recording from RS cells in voltage-clamp mode simultaneously with the LFP in the same barrel. An example recording is shown in *Figure 5D*, with the averaged voltage-clamped traces from the RS neuron (blue) below the averaged LFP (magenta). The vertical guidelines demonstrate that the troughs of the ripplet waveform were aligned with the onset of EPSCs in the RS cell. Plotting ripplet troughs against EPSC onsets in eight RS cells recorded simultaneously with the LFP (in eight slices from four animals) demonstrated the close temporal correspondence between the extracellular troughs and the onsets of intracellular EPSCs (*Figure 5E*), suggesting that population EPSCs in RS cells were a major contributor to the current sinks underlying the LFP transients.

## Discussion

We describe here transient (<25 ms), ultrafast (~400 Hz) network oscillations, which we refer to as 'ripplets,' elicited in thalamorecipient cortical layers by brief (as short as 1 ms) optogenetic stimulation of thalamocortical axons and terminals. The oscillation was reflected in the extracellular voltage (the LFP) as 2–5 negative waves that immediately followed the initial current sink generated by the thalamocortical volley but were dependent on postsynaptic activity within L4 (*Figure 1*). In whole-cell intracellular recordings, inhibitory FS cells fired a highly reproducible burst of 3–6 spikes at the ripplet frequency (*Figure 2*), with spike peaks aligned with LFP troughs (*Figure 4A and C*). As revealed by pairwise recordings, FS cells fired in exquisitely precise, sub-millisecond synchrony (*Figure 3*). Excitatory RS cells mostly fired 1–2 spikes per ripplet, which, as a population, were temporally offset from FS spikes, preceding each FS spike (except the first) by ~1.5 ms and aligned with ripplet peaks (*Figure 4B and C*). Voltage-clamp recordings at an intermediate holding potential revealed a precise, alternating sequence of EPSCs and IPSCs in RS cells (*Figure 5A and B*), phase-locked to ripplets (*Figure 5D*) and synchronized between neighboring RS cells (*Figure 5—figure supplement 1*). On average, each FS spike followed an RS EPSC at short (0.4 ms) latency and preceded an RS IPSC by 0.8 ms (*Figure 5C*), suggesting that FS spikes were elicited by EPSPs shared with the excitatory cells, and in turn evoked monosynaptic IPSPs in excitatory cells.

Taken together, these observations suggest the following synaptic mechanisms for ripplet generation (*Figure 6*). The highly synchronous, optogenetically evoked thalamocortical volley set in motion a cascade of 'spike packets' in subsets of L4 excitatory cells, each packet triggering the next one (*Figure 6A*), similar to feedforward 'synfire chains' (*Abeles, 1991*; *Diesmann et al., 1999*; *Mehring et al., 2003*; *Barral et al., 2019*) except that a given excitatory cell could fire in more than one packet. Although only a subset of all RS cells fired on any given cycle, the strength, reliability, and high prevalence of excitatory RS-RS synapses within L4 barrels (*Feldmeyer et al., 1999*; *Petersen and Sakmann, 2000*; *Schubert et al., 2003*; *Lefort et al., 2009*; *Kameda et al., 2012*; *Koelbl et al., 2015*) ensured that this subset was sufficient to extend the oscillation by one more cycle. The 2.4 ms period of the oscillation was determined by the interval between packets, which was the sum of the synaptic delay (from RS spike to EPSP onset in a postsynaptic RS cell) and the rise time (from EPSP onset to spike in the postsynaptic RS cell). The initial thalamocortical volley, as well as each subsequent RS spike packet, elicited highly synchronous FS spike volleys, supported by the strong TC→FS and RS→FS connectivity in L4 of barrel cortex (*Simons and Carvell, 1989*; *Agmon and Connors, 1992*; *Swadlow et al., 1998*; *Gibson et al., 1999*; *Galarreta and Hestrin, 2001*; *Porter et al., 2001*; *Bruno and Simons, 2002*; *Beierlein et al., 2003*; *Gabernet et al., 2005*; *Bruno and Sakmann, 2006*; *Inoue and Imoto, 2006*; *Sun et al., 2006*; *Cruikshank et al., 2007*; *Kameda et al., 2012*; *Hu and Agmon, 2016*; *Yu et al., 2019*). The delay from an RS spike volley to the evoked FS spike was ~1.5 ms, significantly shorter than the oscillation period, because synaptic latencies, EPSC rise times, and spike rise times are all faster in FS cells (*Hu et al., 2014*; *Tremblay et al., 2016*). This left a gap of 0.9 ms between the FS spike volley and the next RS volley, just enough time for FS cells to elicit a wavefront of feedforward IPSPs in RS cells (*Figure 6B*). This inhibitory wavefront coincided with the rising phase of the RS spikes, too late to block the earliest spikes but just in time to preempt any laggard RS cells from firing. In this manner, FS cells enforced RS firing synchrony and thereby also FS spike synchrony in the next cycle. Only a fraction of all excitatory cells escaped this feedforward inhibitory control and fired on any given cycle, enough to elicit the next cycle but not enough for the excitatory feedforward cascade to turn into a paroxysmal chain reaction. The number of excitatory cells firing likely declined with each cycle, possibly because high-frequency activity suppresses excitatory synapses more than inhibitory ones (*Galarreta and Hestrin, 1998*), until the oscillation died out.

According to this model, both phasic inhibition and phasic excitation, reverberating at the same ultrafast frequency but in antiphase with each other (*Figure 6B*), are necessary for generating this network oscillation. While FS cells are a critical component of the circuit, they do not act as independent pacemakers, and direct FS-FS interactions, chemical or electrical, may not be a necessary feature (see below). Several aspects of this model are at odds with currently proposed models of hippocampal ripples (see below), and this could reflect real differences between the two types of oscillations. Nevertheless, the strong phenomenological similarities between neocortical ripplets and hippocampal ripples (see below) suggest that optogenetically evoked ripplets may be a uniquely accessible model system for deciphering the cellular, synaptic, and network mechanisms underlying other fast and ultrafast neocortical and hippocampal oscillations, mechanisms that to date remain elusive.

## Previous observations of ripplet-like oscillations in the neocortex

Although not nearly as widely reported or intensively studied as hippocampal ripples, neocortical ripplet-like oscillations have been observed previously in both humans and animals, either in response to electrical stimulation of peripheral nerves, or in response to strong, punctate sensory stimuli that are likely to activate afferents in a highly synchronous manner. In humans, noninvasive EEG and MEG scalp recordings revealed fast, transient oscillations (~600 Hz, 10–15 ms duration) in response to electrical stimulation of peripheral nerves, riding over the N20 peak, which reflects the monosynaptic cortical response (reviewed in *Curio, 2000*; *Hashimoto, 2000*). Multi-electrode MEG studies localized the source of these fast oscillations to the primary somatosensory cortex (*Hashimoto et al., 1996*). In a combined EEG and single-unit recording study in awake monkeys, 600 Hz LFP oscillations were found to occur in phase with bursts or single spikes in putative pyramidal cells, in response to electrical stimulation of the median nerve; spike bursts were also observed in response to tactile stimulation (tap) of the finger or hand (*Baker et al., 2003*). Similar MEG oscillations were observed in piglets, in response to electrical stimulation of the snout, and were shown to follow at short latency

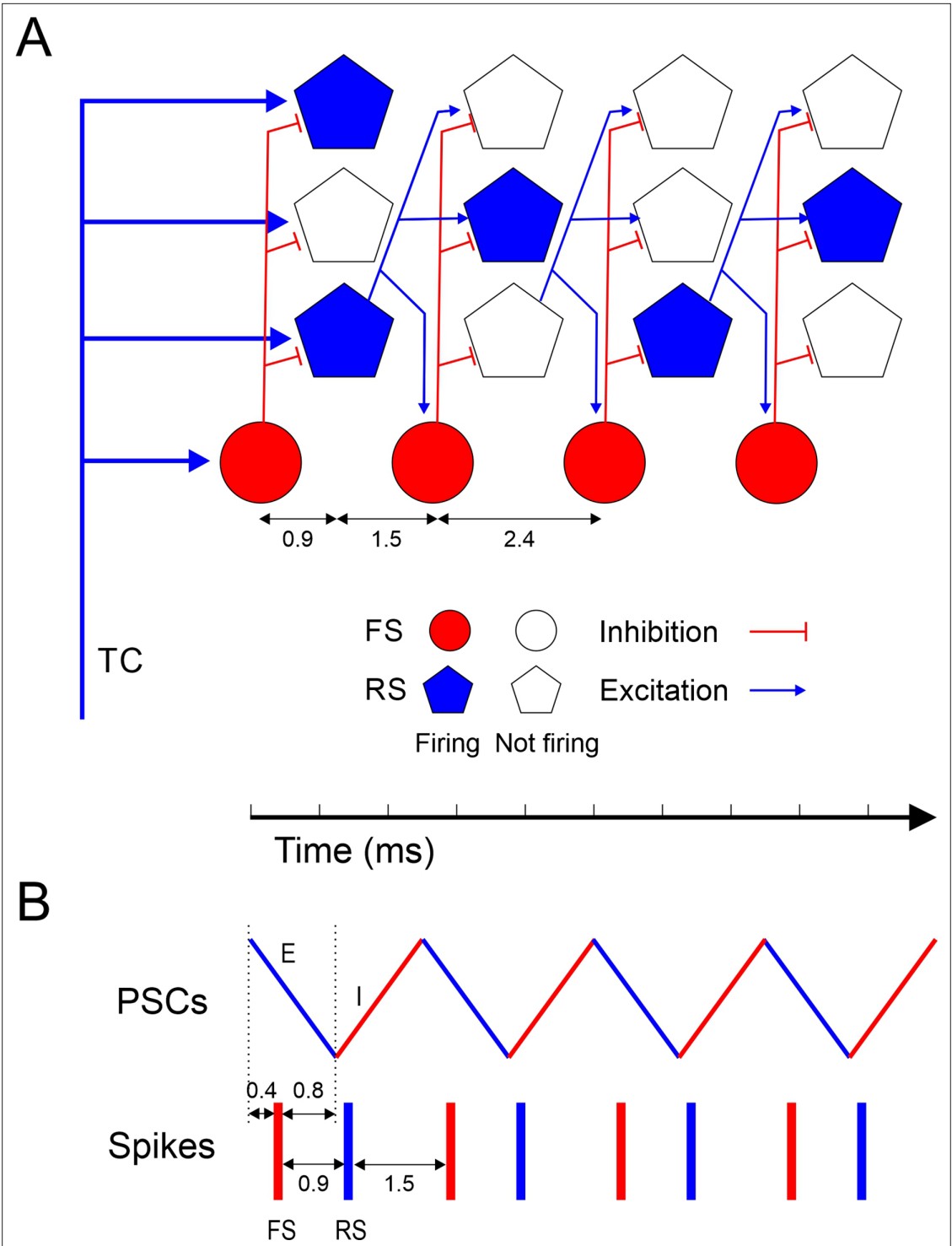

**Figure 6.** Model of ripplet generation. (**A**) Diagram of the layer 4 circuit, illustrating one representative fast-spiking (FS) interneuron and three regular spiking (RS) cells at four successive time points, from left to right, representing the four cycles of the oscillation. Filled symbols represent cells firing, and their position along the horizontal axis represent spike time; hollow symbols represent cells that remain subthreshold. For clarity, only connections from one RS cell are depicted. A synchronous thalamocortical (TC) volley excites FS and RS cells in parallel; FS cells fire ~0.9 ms earlier because of the faster kinetics of their excitatory postsynaptic currents (EPSCs) and spikes. RS cells generate an excitatory feedforward cascade of successive spike volleys that elicit additional FS spike volleys. While FS cells fire on every cycle, RS cells typically fire on only 1–2 cycles per ripplet. The cycle period (2.4 ms) is the sum of the synaptic delay and the postsynaptic time-to-spike in RS cells. (**B**) A schematic timeline of the major events during a ripplet, including FS spikes (red bars), RS spikes (blue bars), and EPSC-IPSC sequences in RS cells (zig-zag line, E's and I's represented by blue and red segments,

*Figure 6 continued on next page*

*Figure 6 continued*

respectively.) FS cells receive EPSPs shared with RS cells; they fire 0.4 ms after EPSP onset and, in turn, elicit IPSPs in RS cells after a 0.8 ms delay, resulting in coincidence of IPSP arrival with onset of spikes in RS cells, enforcing firing synchrony on RS cells.

thalamocortical volleys recorded in cortical layer 4 (*Ikeda et al., 2002*). Electrical stimulation of the thalamus elicited 400–600 Hz LFP oscillations in cortical layers 4–6 in the rat, in phase with extracellular RS and FS spikes and bursts (*Kandel and Buzsáki, 1997*). Probably most reminiscent of the current study, Barth and coworkers observed brief (~13 ms) periods of 350–400 Hz EEG oscillations in barrel cortex of both awake and ketamine-anesthetized rats, in response to short, rapid-onset displacements of single or multiple whiskers; similar oscillations were also observed in auditory cortex in response to sound clicks (*Jones and Barth, 1999*). In a follow-up intracellular study (*Jones et al., 2000*), the same authors report ~500 Hz FS spike bursts in response to whisker deflections, with individual spikes aligned to the troughs in the extracellular field potential. RS cells typically fired 1–2 spikes interspersed with subthreshold events, which occurred at the frequency of the field potential oscillations, precisely as we observed here.

That ripplet-like oscillations have been observed in vivo in response to both sensory and peripheral nerve stimulation, in two primary cortical areas (somatosensory and auditory) and in four different species (humans, monkey, pig, rat), supports the conclusion that the ripplets we observed in yet another species, the mouse, were not an artifact of our ex vivo preparation or optogenetic stimulation, but a manifestation of a ubiquitous intrinsic property of sensory cortices – their propensity for generating ultrafast oscillations in response to strong, synchronous activation of their afferents. In our experiments, the light pulse likely activated with a high degree of synchrony most, if not all thalamocortical terminals impinging on the recorded neurons. Strong synchrony is a well-documented feature of the sensory responses of thalamocortical neurons in vivo, and the degree of thalamic synchrony may be the most important determinant of the strength of the downstream cortical response (*Alonso et al., 1996*; *Pinto et al., 2000*; *Roy and Alloway, 2001*; *Temereanca and Simons, 2003*; *Bruno and Sakmann, 2006*; *Wang et al., 2010*; *Bale et al., 2015*; *Whitmire et al., 2016*; *Wright et al., 2021*). Precisely what degree of thalamocortical synchrony is required for ripplet generation, and what type and intensity of sensory stimuli elicit such synchrony in vivo, is yet to be determined.

If ripplets are an intrinsic feature of sensory neocortex, why have they not been more widely reported in vivo? A technical reason could be the sensitivity of these oscillations to inhalation anesthesia (*Jones and Barth, 1999*), but it could also reflect the fact that most in vivo studies do not record or report LFPs. Without LFP recordings at a high cut-off frequency, ripplets may only be evident in extracellular recordings as bursts in putative FS units, and only if such units are identified and separated from the majority RS units. Such FS bursts have indeed been observed in unanesthetized animals. In awake rabbits, putative FS cells fire ultrafast (>600 Hz) bursts of 3–7 spikes in response to electrical stimulation of the thalamus (*Swadlow, 1995*), reminiscent of the FS bursts we describe here. FS cells in sedated rats fire, on average, four spikes/stimulus in response to high-velocity whisker deflections (*Lee and Simons, 2004*). In awake, head-fixed mice engaged in an object localization task, FS cells often fire short-latency, high-frequency bursts following active whisker touch (*Yu et al., 2016*; *Yu et al., 2019*). In these and similar studies, high-frequency FS bursts may have been embedded in LFP oscillations that were not recorded and therefore remained unrecognized.

## Neocortical ripplets vs. hippocampal ripples

There are many similarities but also a few notable differences between the ripplet events described here and ripples described in the CA1 and CA3 fields of the hippocampus (comprehensively reviewed in *Buzsáki, 2015*). Similar to CA1 pyramidal cells during ripples, individual excitatory cells did not fire on every ripplet cycle, although in our slices the majority of L4 RS cells fired at least one spike per ripplet event, whereas only about 10% of CA1 pyramidal cells fire during a given ripple event in vivo (*Buzsáki et al., 1992*; *Ylinen et al., 1995*), and even a smaller percentage fire ex vivo (*Ellender et al., 2010*; *Hájos et al., 2013*; *Hofer et al., 2015*). The subthreshold E-I sequences we observed in RS neurons (in both current clamp, *Figure 4B*, and voltage clamp, *Figure 5A, B and D*) resembled ripple-locked membrane potential fluctuations observed in hippocampal pyramidal cells in behaving mice (*English et al., 2014*) or antiphasic EPSCs and IPSCs observed in CA1 and CA3 brain slices (*Maier et al., 2011*; *Hájos et al., 2013*; *Schlingloff et al., 2014*). Also similarly to ripples in hippocampus

(*Ylinen et al., 1995*; *Csicsvari et al., 1999b*; *Klausberger et al., 2004*; *Hájos et al., 2013*), excitatory L4 cells fired at or near the negative peak of the ripplet oscillation, while FS interneurons fired ~1.5 ms later. There is also similarity in the triggering mechanism, which in the hippocampus is a volley or burst of volleys in the main excitatory input to CA1, the Schafer collaterals, whereas ripplets were elicited by a spike volley in the main input to the neocortex, thalamocortical axons. Moreover, in the hippocampus the triggering input does not impose its own burst frequency on the target and only acts as a source of strong excitation, and the oscillation is generated de novo in the target structure (*Sullivan et al., 2011*), just as we observed here (*Figure 1G*). Lastly, ripples in ex vivo hippocampus consist of ~4 cycles/event (*Maier et al., 2003*; *Hájos et al., 2013*), similar to ripplets.

The above phenomenological similarities notwithstanding, there was one conspicuous difference between ripplets and hippocampal ripples – their frequency, which was ~400 Hz for ripplets under our ex vivo recording conditions, at least twice as fast as hippocampal ripples in vivo (*Buzsáki et al., 1992*) or in brain slices (*Maier et al., 2003*; *Ellender et al., 2010*). Whether this difference reflects quantitative variations in the detailed cellular and synaptic properties of each circuit, or qualitatively different mechanisms altogether, remains to be determined.

## Mechanisms of ripple and ripplet generation

Although widely studied, the mechanisms pacing hippocampal ripples are still under debate (reviewed in *Buzsáki, 2015*). Several competing models are proposed in the literature, assigning different weights to pyramidal–pyramidal (PYR-PYR), pyramidal–interneuron (PYR-INT), and interneuron–interneuron (INT-INT) connections (*Traub and Bibbig, 2000*; *Taxidis et al., 2012*; *Traub et al., 2012*; *Malerba et al., 2016*; *Donoso et al., 2018*; *Braun and Memmesheimer, 2022*). Recent experimental and computational studies lean in favor of a 'hybrid' PYR-INT-INT model, suggesting that FS interneurons act as the central pacemakers of the ripple oscillation, and that tonic excitation from pyramidal neurons is needed only to keep interneurons sufficiently depolarized (*Schlingloff et al., 2014*; *Stark et al., 2014*; *Gan et al., 2017*; *Ramirez-Villegas et al., 2018*). In contrast, we propose that neocortical ripplet oscillations were based on what may be considered a PYR-PYR-INT mechanism, in that phasic excitatory inputs originating from local RS cells were driving the oscillation on a cycle-by-cycle basis, and that FS interneurons were followers, rather than leaders, in these oscillations. This conclusion is based on our voltage-clamp data that revealed regularly alternating, ripplet-entrained postsynaptic E-I sequences in RS cells (*Figure 5A, B and D*). Our data also showed that EPSCs in RS cells immediately preceded spikes in simultaneously recorded FS cells (*Figure 5C*), implying that FS firing was driven by rhythmic excitatory inputs shared with the RS cells and not by tonic depolarization. In turn, FS spike volleys generated antiphasic IPSCs in RS cells, and this precisely timed inhibition was likely critical for maintaining synchronous RS (and thereby also FS) firing from cycle to cycle. Notably, similar sequences of antiphasic, ripple-locked EPSCs and IPSCs were also observed in CA1 and CA3 pyramidal neurons ex vivo (*Maier et al., 2011*; *Hájos et al., 2013*; *Schlingloff et al., 2014*; but see *Gan et al., 2017*). Indeed, it is difficult to see how such precisely timed, phasic excitation would *not* be a driving factor in the entrainment of FS spikes to the ripplet (or ripple) oscillation, even if additional, interneuron-autonomous mechanisms (see next section) contributed to FS-FS synchrony.

## The role of electrical and chemical FS-FS connections

FS cells in the rodent neocortex and hippocampus are known to be coupled by electrical synapses. These are prominent in juvenile rats, with coupling coefficients of 5–10% (*Galarreta and Hestrin, 1999*; *Gibson et al., 1999*; *Gibson et al., 2005*; *Otsuka and Kawaguchi, 2013*). In the mouse, however, electrical coupling is much weaker, with reported coupling coefficients of ~1.5% in juvenile (*Hu et al., 2011*; *Hu and Agmon, 2015*) and adult (*Galarreta and Hestrin, 1999*; *Galarreta and Hestrin, 2002*; *Meyer et al., 2002*) mice, although one study in juvenile mice *Deans et al., 2001* found stronger coupling. In our current dataset of subadult and adult animals, the coupling coefficient between paired L4 FS cells, when detectable, was <1%. Interestingly, a recent ultrastructural study of gap junctions in barrel cortex has identified a subset of PV interneurons ('Type 1'), with somata located more centrally in the barrel and dendritic arbors fully contained within the barrel, which do not make gap junctions with each other but do make them with PV cells closer to the barrel periphery (*Shigematsu et al., 2019*). It is possible that our paired recordings, which were biased towards neighboring FS cells, preferentially sampled unconnected type 1 pairs. Even so, the pairs we

recorded were exquisitely synchronized. Our data therefore suggest that electrical coupling did not play a major role in the submillisecond FS-FS synchrony we observed and by extension in the generation of ripplets. This conclusion is consistent with previous studies in connexin 36 (Cx36) knockout mice, in which FS-FS electrical coupling is abolished but hippocampal ripples are mostly unaffected (*Hormuzdi et al., 2001*; *Maier et al., 2002*; *Buhl et al., 2003*). Notwithstanding, we cannot rule out that dendro-dendritic gap junctions, which have been described between PV-expressing interneurons (*Katsumaru et al., 1988*; *Fukuda and Kosaka, 2000*; *Fukuda and Kosaka, 2003*; *Fukuda, 2017*) and which may not be evident in somatic recordings, played a supporting role in synchronizing FS spiking during ripplets, in conjunction with GABAergic synapses (*Gibson et al., 1999*; *Tamás et al., 2000*; *Szabadics et al., 2001*; *Hu and Agmon, 2015*) or with shared, synchronous EPSPs (*Galarreta and Hestrin, 2001*; *Hjorth et al., 2009*).

In contrast to the sparsity of electrical coupling, FS-FS cells in the mouse and rat cortex are robustly connected chemically by one-way or reciprocal GABAergic synapses (*Gibson et al., 1999*; *Deans et al., 2001*; *Galarreta and Hestrin, 2002*; *Gibson et al., 2005*; *Ma et al., 2012*; *Hioki et al., 2013*; *Pfeffer et al., 2013*), and such chemical coupling can synchronize neurons with submillisecond precision (*Gibson et al., 2005*; *Hu et al., 2011*). In our current dataset, only 1/3 of the recorded FS-FS pairs were connected chemically, whereas all pairs exhibited sub-millisecond firing synchrony during ripplets (*Figure 3*). This argues against a major role for inhibitory-to-inhibitory synapses in this synchrony, although we cannot rule out the possibility that such synchrony is transitive; that is, if A and B are synchronized by mutual inhibition, and so are B and C, then A and C will be synchronized even if they are not directly connected. Nevertheless, the most likely generator of submillisecond FS-FS synchrony during ripplets remain shared excitatory inputs. Similar conclusions were reached about sharp synchrony between FS interneurons in the awake rabbit (*Swadlow et al., 1998*) and during cortical UP states in mice (*Neske and Connors, 2016*).

## Ideas and speculations: The functional significance of ripplets

What function do ripplets serve in cortical sensory processing, if any? One possibility is that ripplets signal in a binary mode, 'Yes' or 'No,' that a highly salient sensory event has occurred. In addition, the spike sequences of excitatory cells during ripplets could provide increased bandwidth for transmission and storage of specific information about this event. Similar to the encoding scheme proposed for gamma/theta oscillations (*Lisman and Buzsáki, 2008*; *Lisman and Jensen, 2013*) or for odor coding in the insect olfactory system (*Laurent and Davidowitz, 1994*), ripplets could provide a temporal framework for clustering spikes into discrete time bins, each ripplet cycle representing a time bin of ~2.5 ms. While FS spikes occur in every bin, excitatory neurons fire more sparsely, thus generating a binary sequence of 1's (spike) and 0's (no spike), with different neurons generating different sequences. Moreover, these binary sequences may vary in tune with the temporal (*Arabzadeh et al., 2005*) or spatial (*Andermann et al., 2004*) variations in the activity pattern of primary sensory, brainstem, and thalamocortical neurons along the chain of activity from periphery to cortex, thus encoding specific information about the sensory event. The assignment of a precise spike sequence to a neuron, rather than merely a total spike count, more than doubles the amount of information that can be conveyed by the neuron (*Arabzadeh et al., 2006*). The amount of information depends on the temporal resolution of the spike code, and it is therefore quite striking that three different studies of temporal coding in rat somatosensory cortex (*Ghazanfar et al., 2000*; *Panzeri et al., 2001*; *Foffani et al., 2004*) found that maximal information content is achieved when spikes are aggregated in time bins of 2–3 ms, which is exactly a ripplet cycle.

How will the information encoded in these putative binary patterns be read-out by downstream cortical targets? L4 excitatory cells make synapses on basal dendrites of L2/3 pyramidal neurons (*Feldmeyer et al., 2002*; *Lübke et al., 2003*; *Shepherd and Svoboda, 2005*; *Lefort et al., 2009*), and at least some L2/3 pyramidal neurons are tuned to precise spatiotemporal sequences of synapse activation (*Branco et al., 2010*; *Branco and Häusser, 2011*). One can therefore envision that a given spike pattern in L4 cells will specifically activate those L2/3 pyramidal cells that are tuned to this exact spatiotemporal pattern of synaptic inputs. In this manner, ripplets can transform a temporal code representing a sensory feature, for example, texture, into a spatial code, with different ensembles of L2/3 pyramidal cells encoding different textures. Such ensembles can stabilize by Hebbian or spike time-dependent plasticity mechanisms and form 'engrams,' long-term assemblies that can later be

activated to recall the same sensory events that initially shaped them (*Liu et al., 2012*; *Carrillo-Reid et al., 2019*). That ripplets participate in the encoding of sensory information into transient or stable ensembles is a testable hypothesis.

## Summary and future directions

Our experiments revealed the synaptic interactions underlying ripplets – transient, ultrafast network oscillations evoked in the somatosensory cortex by a brief, synchronous thalamocortical volley. Previously observed in human EEG/MEG recordings and in a small number of in vivo animal studies, ripplet-like oscillations in sensory cortices may be much more common and widespread than currently recognized. The robustness and precise reproducibility of optogenetically evoked ripplets, and the accessibility of our ex vivo experimental model to recording and imaging and to pharmacological, chemogenetic, and optogenetic manipulations, will allow future studies to examine the cellular and synaptic mechanisms of ripplets in exquisite detail, shedding light not only on this particular phenomenon but also on the synaptic circuitry of the thalamorecipient cortical layers in general. This will accelerate development of detailed, biologically grounded computational models of the neocortical network; indeed, the ability to generate ripplet oscillations in response to a synchronous thalamocortical volley can now be used as a stringent 'reality check' for existing models (*Banitt et al., 2007*; *Markram et al., 2015*; *Hass et al., 2016*; *Gouwens et al., 2018*; *Borges et al., 2022*). Studies of neocortical ripplets are also likely to shed new light on the synaptic mechanisms underlying similar oscillations in other brain areas, such as hippocampal ripples. Beyond mechanisms, the role of ripplets in encoding and transmitting sensory information is a highly intriguing topic awaiting future investigations.

## Materials and methods

**Key resources table**

| Reagent type (species) or resource | Designation | Source or reference | Identifiers | Additional information |
|---|---|---|---|---|
| Strain, strain background (mouse, *Mus musculus*, both sexes) | KN282 | Mutant Mouse Resource and Research Center (MMRRC) | 036680-UCD | http://www.gensat.org/cre.jsp |
| Strain, strain background (mouse, *M. musculus*, both sexes) | Ai32 | The Jackson Laboratory | RRID:IMSR_JAX: 024109 | https://www.jax.org/strain/024109 |
| Strain, strain background (mouse, *M. musculus*, both sexes) | Ai9 | The Jackson Laboratory | RRID:IMSR_JAX: 007909 | https://www.jax.org/strain/007909 |
| Chemical compound, drug | CNQX | Tocris | Cat. no. 1045/1 | 20 µM |
| Chemical compound, drug | D-AP5 (APV) | Tocris | Cat. no. 0106/1 | 30 µM |
| Chemical compound, drug | Gabazine (SR 95531 hydrobromide) | Tocris | Cat. no. 1262/10 | 20 µM |

## Animals

Animals used in this study were housed at the AAALAC-accredited WVU Lab Animal Research Facility according to institutional, federal, and AAALAC guidelines. Animal euthanasia for brain slice preparation and for perfusion fixation was carried out under deep anesthesia. Animal use followed the Public Health Service Policy on Humane Care and Use of Laboratory Animals and was approved by the WVU Institutional Animal Care and Use Committee (protocol #1604002316). West Virginia University has a PHS-approved Animal Welfare Assurance D16-00362 (A3597-01).

We crossed KN282 mice (MMRRC strain 036680-UCD), in which neurons in the ventrobasal (VB) complex of the thalamus express Cre recombinase (*Gerfen et al., 2013*; *Hu and Agmon, 2016*), with the Ai32 channelrhodopsin reporter (strain #012569, The Jackson Laboratory, Bar Harbor, ME) (*Madisen et al., 2012*) or with the Ai9 tdTomato reporter (strain #007909) (*Madisen et al., 2010*). Dual-transgenic progeny of both sexes, 3–7 weeks old, were used for experiments. All mouse lines used were maintained for multiple generations as heterozygotes by breeding mutant males with outbred (CD-1, Charles River Laboratories, USA) wild-type females.

## Slice preparation

Mice were decapitated under deep isoflurane anesthesia, the brains removed and submerged in ice-cold, sucrose-based artificial cerebrospinal fluid (ACSF) containing the following (in mM): sucrose 206,

NaH$_2$PO$_4$ 1.25, MgCl$_2$.6H$_2$O 10, CaCl$_2$ 0.25, KCl 2.5, NaHCO$_3$ 26, and D-glucose 11, pH 7.4. Thalamocortical brain slices (*Agmon and Connors, 1991*; *Porter et al., 2001*) of somatosensory (barrel) cortex, 300–350 µm thick, were cut in same solution using a Leica VT-200 vibratome and placed in a submersion holding chamber filled with recirculated and oxygenated ACSF (in mM: NaCl 126, KCl 3, NaH$_2$PO$_4$ 1.25, CaCl$_2$ 2, MgSO$_4$ 1.3, NaHCO$_3$ 26, and D-glucose 20). Slices were incubated for at least 30 min at 32°C and then at room temperature until use. For recording, individual slices were transferred to a submersion recording chamber and continuously superfused with 32°C oxygenated ACSF at a rate of 2–3 ml/min.

## Electrophysiological recordings

Recording were done on an upright microscope (FN-1, Nikon) under a 40x water immersion objective. For whole-cell recordings, glass micropipettes (typically 5–8 MΩ in resistance) were filled with an intracellular solution containing (in mM): K-gluconate 134, KCl 3.5, CaCl$_2$ 0.1, HEPES 10, EGTA 1.1, Mg-ATP 4, phosphocreatine-Tris 10, and 2 mg/ml biocytin, adjusted to pH 7.25 and 290 mOsm. Layer 4 and 5B FS and RS cells were identified visually by soma size and shape and targeted for single or dual whole-cell recordings using a MultiClamp 700B amplifier (Molecular Devices, San Jose, CA). Upon break-in, cells were routinely tested by a standardized family of incrementing 600-ms-long intracellular current steps in both negative and positive directions. Upon strong depolarization, FS cells invariably reached steady-state firing rates of 150–300 Hz and their spike trains showed no or little frequency adaptation, while RS cells typically fired an initial fast doublet, followed by a slower, steady-state firing at <50 Hz (*Figure 2—figure supplement 1*). In post-hoc analysis, the same records were used to extract eight electrophysiological parameters for each cell (*Figure 2—source data 1*). In dual-recording experiments, synaptic connectivity was routinely tested in both directions by stimulating one cell with a 10–20 Hz train of brief suprathreshold current steps and recording the response in the paired cell (at resting potential if the putative synapse was excitatory, or at –50 mV if inhibitory). Extracellular LFPs were recorded using glass micropipettes filled with 0.9% sodium chloride, with a tip broken under visual control (tip diameter ~5 µm, 2–3 MΩ resistance), positioned in the center of the L4 barrel. The extracellular signal was amplified 1000× and low-pass filtered at 1.3 kHz using a differential amplifier (EXT-02F, NPI, Tamm, Germany); 20–100 trials were averaged. To block fast excitatory synaptic transmission, 6-cyano-7-nitroquinoxaline-2,3-dione disodium (CNQX, Tocris, Minneapolis, MN) and d-(−)–2-amino-5-phosphonopentanoic acid (D-APV, Tocris) were added to the ACSF at final concentrations of 20 and 30 µM, respectively. To block fast inhibition locally, a patch pipette filled with a solution of 20 µM Gabazine (Tocris) in ACSF was inserted superficially into the slice adjacent to the recording pipette; drug was then infused locally by applying positive pressure through the pipette shank. Data were acquired at a 20 kHz sampling rate using a National Instruments (Austin, TX) ADC board and in-house acquisition software written in the LabView (National Instruments) environment. Reported intracellular voltages are not corrected for liquid junction potential.

## Data inclusion/exclusion

Intracellular recordings not passing our minimal quality criteria (resting potential ≤–60 mV [FS cells] or ≤–65 mV [RS cells] without holding current, action potential peak ≥10 mV [FS cells] or ≥20 mV [RS cells]), were excluded from further analysis and reporting. Additional inclusion criteria for each analysis are stated in the 'Results' section. No datapoints (spikes, cells, cell pairs, or slices) within the stated inclusion criteria were excluded from the analysis, figures, or reported results.

## Optogenetic stimulation

To activate thalamocortical axons and terminals, pulses of white light from a high-power LED light source (Prizmatix, Holon, Israel) were delivered through the epi-illumination light path and a GFP filter cube. Maximal illumination intensity of the Prizmatix light source was 1.0 mW, as measured in air at the focal plane of the 40x objective. Assuming the full light energy was spread evenly over the field of view of the 40x objective (550 µm diameter) and no light scattering in water, the estimated maximal illuminance at the tissue was 4.2 mW/mm$^2$. In the course of the experiment, light intensity was adjusted to 10–90% of maximal intensity. Stimuli at each light level were typically repeated 8–12 times at 6–10 s intervals.

## Synchrony analysis

Given any two spike trains, synchrony is defined as the fraction of spikes in the shorter train occurring within a predetermined 'synchrony window' (SW) before or after a spike in the longer train. Chance synchrony is the expected (i.e. average) synchrony remaining after shifting each spike in the shorter train by a random jitter ≤J, J being the 'jitter window,' and is calculated analytically rather than by Monte Carlo simulations. The JBSI is the difference between the observed and chance synchrony, that is, the magnitude of synchrony destroyed by the jitter, multiplied by a normalization factor β which limits the JBSI to values between 1 (highest possible synchrony) to -1 (lowest possible synchrony), with 0 indicating purely chance synchrony. Here we used $J = 2 \cdot SW$ and $\beta = 2$. Unlike many other synchrony measures, the JBSI is independent of firing rates and firing rate differentials, and is not sensitive to slow co-modulations in these rates. See *Agmon, 2012* for theoretical and computational details. JBSI computations were done using in-house MathCad routines (MathCad, PTC). The routines, as well as an equivalent MATLAB function, are available under the MIT license on https://github.com/aricagmon/JBSI-codes (copy archived at *Agmon, 2022*).

## Statistical analysis

Unless noted otherwise, exact p-values were computed using distribution-free, non-parametric permutation tests by performing 10,000 random permutations of the data and calculating the fraction of permutations resulting in equal or more extreme values of the relevant statistic under both tails (*Good, 1999*). When no more extreme values were found, this is indicated as $p < 0.0001$. All data are reported as mean ± SEM unless indicated otherwise.

## Acknowledgements

We thank Qingyan Wang for outstanding technical support. We thank Barry Connors and Liset Menendez de la Prida for critical reading of the manuscript and helpful discussions. Craig Atencio translated the JBSI algorithm into a MATLAB code. This study was supported by the National Institutes of Health grant NS116604 to AA. Additional funding was provided by Transition Grant Support from the Office of Research and Graduate Education, WVU Health Sciences Center, and by the Program to Stimulate Competitive Research, Office of Research, WVU. REH was supported by NIH training grants GM081741 and GM132494. Confocal imaging was performed at the WVU Microscopic Imaging Facility, which was supported by NIH grants GM121322, GM103434, GM103503, GM104942, and RR016440.

## Additional information

### Funding

| Funder | Grant reference number | Author |
|---|---|---|
| National Institutes of Health | NS116604 | Ariel Agmon |
| National Institutes of Health | Predoctoral Training Grants GM081741 and GM132494 | Rachel E Hostetler |

The funders had no role in study design, data collection and interpretation, or the decision to submit the work for publication.

### Author contributions

Hang Hu, Formal analysis, Investigation, Writing - review and editing, Electrophysiological recording and analysis; Rachel E Hostetler, Formal analysis, Investigation, Writing - review and editing, Histology, confocal imaging, image analysis; Ariel Agmon, Conceptualization, Formal analysis, Funding acquisition, Writing - original draft, Writing - review and editing

### Author ORCIDs

Rachel E Hostetler http://orcid.org/0000-0002-4185-6468
Ariel Agmon http://orcid.org/0000-0002-7556-8130

### Ethics

Animals used in this study were housed at the AAALAC-accredited WVU Lab Animal Research Facility according to institutional, federal and AAALAC guidelines. Animal use followed the Public Health Service Policy on Humane Care and Use of Laboratory Animals, and was approved by the WVU Institutional Animal Care and Use Committee (protocol #1604002316). West Virginia University has a PHS-approved Animal Welfare Assurance D16-00362 (A3597-01).

### Decision letter and Author response

Decision letter https://doi.org/10.7554/eLife.82412.sa1
Author response https://doi.org/10.7554/eLife.82412.sa2

## Additional files

### Supplementary files
• MDAR checklist

### Data availability

Figure 1- Source Data 1 contains the cell count data used for Figure 1 - Figure Supplement 1; Figure 2- Source Data 1 contains the electrophysiological parameters data used for Figure 2 - Figure Supplement 1;Code used to calculate synchrony indices has been deposited to GitHub.

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
