## [Editor Report]

This study examines the potential neuronal basis for generating ultrafast oscillations (250-600Hz) in the cortex evoked by optogenetic stimulation of thalamocortical afferents in ex vivo brain slices. The authors proposed that these oscillations correlate with sensory stimulation and may be relevant for the perception of relevant sensory inputs and they combined ex-vivo whole-cell patch-clamp recordings, local field potential (LFP) recordings, and optogenetic activation of thalamic afferents to generate ripple-like extracellular waveforms in the cortex, referred to as "ripplets." The authors described the sequences of RS and FS neuron discharge and how they phase-locked to the ripplet, providing a model for the cellular mechanism generating the ripplet. The authors also cited the literature about ultrafast oscillations and carefully compared the novel ripplets to the well-known hippocampal ripples.

---

## [Decision Letter]

**Decision letter after peer review:**

Thank you for submitting your article "Ultrafast (>400 Hz) network oscillations induced in thalamorecipient cortical layers by optogenetic activation of thalamocortical axons" for consideration by *eLife*. Your article has been reviewed by 3 peer reviewers, one of whom is a member of our Board of Reviewing Editors, and the evaluation has been overseen by Laura Colgin as the Senior Editor. The following individuals involved in the review of your submission have agreed to reveal their identity: John R Huguenard (Reviewer #2); Laszlo Acsady (Reviewer #3).

Essential revisions:

Based on the attached reviewers' comments, all the reviewers agree that this manuscript is of potential interest to the audience studying the mechanisms of high-frequency cortical oscillations and their role in sensory information transfer. All three reviewers also agree that the proposed method provides a successful ex vivo model to study the ultrafast oscillation-mediated circuitry and cellular mechanisms and that the data reasonably support most of the claims by the authors in this manuscript.

However, all three reviewers also raised concerns in common about what extent the data could be interpreted based on the present data.

Firstly, they raised the concern that the experiments claiming the spread of phasic activity from Layer 4 RS to Layer 5 FS cells can not be accepted as entirely conclusive as it is at the current state. In more detail, the horizontal cut severed the L4 RS to L5 FS connections and many TC inputs to the L5 FS apical dendrites and the axons of L4 FS cells to L5 FS cells which can be pivotal in the translaminar spread. The observation that L5 FS spike bursts decrease after cutting the L4/L5 boundary could be interpreted in other ways. The current interpretation of data seems overly extended because it does not necessarily represent cellular and synaptic activities that are phase-locked with the ripplets observed in L4. As it is outside the main scope of the paper, it could be toned down or dropped. Alternatively, this part should be redone with appropriate control experiments with the angled slice-cutting methods and the experiments suggested by the reviewers.

The second common concern is the interpretation of nearby FS synchrony induced by TC activation. The authors described the phase-locked responses of FS and RS cells with local ripple oscillations to optogenetic activation and highly synchronized FS-FS firing by eliminating electrical gap-junction and inhibitory synaptic connections to this synchrony. Based on these findings, the authors suggest that common excitatory inputs to FS and RS cells in L4 would be essential to generate these local ripplets. However, it interferes with the ability to follow the logical flow for biding other findings of phase-locking responses of FS and RS cells in ripplet oscillations in L4. This issue may be addressed by rephrasing and stating that Opto activation of TC creates highly synchronized inputs to the L4, which might further help the ultrafast synchrony.

Throughout the manuscript, the authors understated the GABAergic mechanism by FS cells, whereas the highly (and artificial) synchronous stimulation of many TC axons and terminals would activate FS cells which are GABAergic neurons.

Thirdly, as the authors proposed a circuit model, it would be ideal that the authors try to perform in silico analysis using the suggested model to explore the function of thalamocortical axons on the fast-spiking and regular spiking neurons to support their circuit model. However, if it takes too much effort to test an in silico model, it can be considered as an option for the authors.

There are more major concerns raised by the reviewers below, which should be addressed appropriately before we can consider publication in *eLife*.

Should further data analysis, figures revision, and text rationale allow you to address these criticisms, we would be happy to review a revised manuscript.

In the meantime, we hope you will find our referees' comments helpful. Please do not hesitate to contact us if there is anything you would like to discuss.

*Reviewer #1 (Recommendations for the authors):*

1) The authors should consider the logical flow when presenting the synchrony of the FS-FS spike burst. It would strongly support the main idea of the manuscript if the authors rephrase or/and rearrange the corresponding text and data.

2) Please provide detailed information on the viral vector of JAWS used for the experiment of suppressing L4 RS cell firing in the methods section, and also present expression image and spike data in figures to support the idea for the subsequent experiment performed in L5.

*Reviewer #2 (Recommendations for the authors):*

1) While the authors are generally cautious in their interpretation of the optogenetic approach and how realistic it might be, we think that this issue deserves more direct discussion. In some ways, the high degree of nearby FS synchrony induced by TC optical activation is not at all surprising given what is already known about the potency of TC inputs onto FS cells (the senior author's own work and that of many other labs), and the highly (and artificial) synchronous stimulation of many TC axons and terminals onto FS cells. The rare literature examples of high-frequency sensory evoked responses that the authors cite are arguably not simple sensory activity at all (electrical stimulation in the periphery or thalamus, both of which might lead to abnormal sensory recruitment.

2) The results regarding the propagation of high-frequency events to layer 5 are arguably outside the main scope of the paper, and could potentially be dropped. As is, many controls that have not been provided would need to be added. For example, demonstrating a complete lack of optogenetic effects in layer 4 with Jaws to rule out that there might actually be low levels of expression that could contribute to the results. In addition, ruling out the interaction between the blue and red light stimuli in the two pathways is required, as the spectral overlap for excitation spectrum is not as clean as one would need to unambiguously interpret these results.

*Reviewer #3 (Recommendations for the authors):*

1) In order to prove the critical role of FS-FS interactions in timing the oscillation the authors may consider GABA-A receptor blockers and/or optogenetic inhibition of PV cells.

2) To gain further insights into the excitatory background of the oscillation I would propose performing RS-RS pair recordings. In this configuration, the kinetics and strength of their interactions could be measured to establish whether this is fast and strong enough to provide a drive for the oscillation. RS data similar to FS cells shown in Figure 2 (variable stimulation intensity vs jitter) could also help to elucidate the mechanisms. Using voltage-clamp experiments in RS cells could help to resolve the origin of the excitatory drive of the oscillation (TC or RS-RS interactions). Optimally experimental work would be coupled with in silico simulations to test which network activity may underly the oscillation.

3) In order to make predictions for future research concerning the role of this ultrafast oscillation it would be advisable to test its postsynaptic effects on targets that do not receive direct thalamic inputs (i.e. upper layer 2/3 cells). This would not only allow more insights into the generation of the oscillation but could test whether the oscillation is indeed able to select a subset of L2/3 cells as suggested. For a translaminar spread to layer 3 measuring the strength and variability of the oscillation in layer 2/3 pyramidal cells, which are not directly contacted by thalamic inputs, would greatly enhance the predictive power of the paper for future research.

[Editors' note: further revisions were suggested prior to acceptance, as described below.]

Thank you for submitting your revised manuscript titled "Ultrafast (400 Hz) network oscillations induced in middle cortical layers by optogenetic activation of thalamocortical axons." We appreciate the effort you have made to improve your manuscript, and we have carefully reviewed it.

We are pleased to see that the authors have addressed most of the comments made by the reviewers. The manuscript has been improved, and the unnecessary distracting parts have been removed, resulting in a smoother flow of data. However, we would like to draw your attention to a few critical comments that still need to be addressed.

Firstly, the authors should include a discussion on how the strong optogenetic approaches used in this paper might or might not be relevant to the actual sensory processing in the brain circuit, given that the current data is obtained from brain slices and artificially synchronous wide-field optogenetic activation. We believe that this is an essential point to address, and we encourage you to include an appropriate discussion in the manuscript.

Secondly, the authors should discuss in more detail the potential functional consequences and function of ultrafast brain oscillation in vivo, given that this manuscript focuses on the cellular nature of high-frequency oscillation in vitro. This is an excellent suggestion, and we believe that including a discussion on this topic will enhance the impact of your manuscript.

We appreciate the valuable contribution that your manuscript has made, and we believe that addressing these comments on functional significance will further improve the quality and impact of your work. We have appended the full comments from the reviewers below for your reference.

*Reviewer #2 (Recommendations for the authors):*

We agree that this is a carefully controlled study that examines the mechanisms through which very high-frequency LFP activity can be evoked in sensory cortical slices. The recordings are elegant, with combined extracellular and multi-intracellular data, and high quality and the results clearly lead to the conclusion that this in vitro activity is driven by RS cell firing.

However, regarding original major point #1, we do not share the opinion of the authors that these are well linked to naturalistic sensory stimuli. In all the papers cited to support this, the main means of activating cortical response is electrical shocks either delivered directly to peripheral sensory nerves (non-human primates, humans), or to the snout (piglets). Baker found LFP wavelets with electrical shocks to peripheral nerves but was not surprised by this (their last paragraph), as it presumably results from a particular form of artificial activation. They did find multiple spikes (perhaps consistent with LFP wavelets) with tactile sensory responses, but notably did not observe LFP correlates to these more naturalistic stimuli. Jones found wavelets, particularly with simultaneous stimulation of multiple vibrissae, but also looked at more controlled stimuli of multiple whiskers. This last paper is the only one that comes close to reporting cortical high-frequency wavelets with what might be considered natural sensory.

What the authors might consider "exceptionally salient stimuli" (line 72) is a bit unclear, but if by this they mean response to an electrical shock, we would argue this does indeed result from abnormal sensory recruitment, and presumably would not be part of sensory processing in 99.9% of contexts. The issue is that optogenetic stimulation artificially increases synchrony perhaps even more than electrical peripheral stimulation, as all the infected TC axons are presumably rapidly and synchronously activated by the light stimulus. Thus the abstract, and related text in the introduction and discussion needs to be modified to reflect the proper context. The following abstract statement "At the higher end of the scale, ultrafast (400-600 Hz) oscillations in the somatosensory cortex, in response to peripheral stimulation, were previously observed in human and a handful of animal studies;" should be revised to indicate that most of the literature on this point were obtained in fact with responses to electrical shocks, which are not a proxy for generic (naturalistic?) peripheral sensory stimulation, but rather an artificial means of activating sensory pathways.

*Reviewer #3 (Recommendations for the authors):*

The authors adequately answered my queries concerning the mechanism of generating ultrafast network oscillation via the interacting elements of layer 4 neurons following the activation of their thalamic inputs. Now the comparison with earlier hippocampal data is fair, detailed, and concise. I have no more comments.

---

## [Author Response]

Essential revisions:Based on the attached reviewers' comments, all the reviewers agree that this manuscript is of potential interest to the audience studying the mechanisms of high-frequency cortical oscillations and their role in sensory information transfer. All three reviewers also agree that the proposed method provides a successful ex vivo model to study the ultrafast oscillation-mediated circuitry and cellular mechanisms and that the data reasonably support most of the claims by the authors in this manuscript.However, all three reviewers also raised concerns in common about what extent the data could be interpreted based on the present data.Firstly, they raised the concern that the experiments claiming the spread of phasic activity from Layer 4 RS to Layer 5 FS cells can not be accepted as entirely conclusive as it is at the current state. In more detail, the horizontal cut severed the L4 RS to L5 FS connections and many TC inputs to the L5 FS apical dendrites and the axons of L4 FS cells to L5 FS cells which can be pivotal in the translaminar spread. The observation that L5 FS spike bursts decrease after cutting the L4/L5 boundary could be interpreted in other ways. The current interpretation of data seems overly extended because it does not necessarily represent cellular and synaptic activities that are phase-locked with the ripplets observed in L4. As it is outside the main scope of the paper, it could be toned down or dropped. Alternatively, this part should be redone with appropriate control experiments with the angled slice-cutting methods and the experiments suggested by the reviewers.

We agree with the editors and with all three reviewers that the experiments described in the original Figure 6 are not conclusive, as they do not test directly ripplet mechanisms in L4, and as they are open to alternative interpretations. We especially took notice of the comment by Reviewer #3 that the L4/L5 cut necessarily truncated the apical dendrites of L5 pyramidal cells, which could have removed some of their thalamocortical inputs and which could have also rendered these cells less responsive to local excitation. We note that this interpretation is still fully consistent with our model, which posits that firing of local excitatory neurons is necessary for pacing FS interneurons at ripplet frequency. Nevertheless, we felt that more work is required to arrive at an unequivocal interpretation of these experiments, and removed the slice cutting experiment from the current manuscript. We still report the observation that L5 FS interneurons fire light-induced bursts at 400 Hz, and provide examples in Figure 2 —figure supplement 2.

The second common concern is the interpretation of nearby FS synchrony induced by TC activation. The authors described the phase-locked responses of FS and RS cells with local ripple oscillations to optogenetic activation and highly synchronized FS-FS firing by eliminating electrical gap-junction and inhibitory synaptic connections to this synchrony. Based on these findings, the authors suggest that common excitatory inputs to FS and RS cells in L4 would be essential to generate these local ripplets. However, it interferes with the ability to follow the logical flow for biding other findings of phase-locking responses of FS and RS cells in ripplet oscillations in L4. This issue may be addressed by rephrasing and stating that Opto activation of TC creates highly synchronized inputs to the L4, which might further help the ultrafast synchrony.Throughout the manuscript, the authors understated the GABAergic mechanism by FS cells, whereas the highly (and artificial) synchronous stimulation of many TC axons and terminals would activate FS cells which are GABAergic neurons.

This recommendation appear to be based on comments by Reviewer #1, who was concerned about the flow of logic in the manuscript, with the FS-FS synchrony seemingly disconnected from the other findings. To address this concern we switched the order of the original Figure 3 and Figure 4, so the FS-FS synchrony is now described immediately after the description of FS spike bursts in Figure 2. The logic of examining FS-FS synchrony is also better explained now.

Thirdly, as the authors proposed a circuit model, it would be ideal that the authors try to perform in silico analysis using the suggested model to explore the function of thalamocortical axons on the fast-spiking and regular spiking neurons to support their circuit model. However, if it takes too much effort to test an in silico model, it can be considered as an option for the authors.

We agree that a computational model of ripplets would be very informative – not only for deciding between competing mechanisms, but also for elucidating the parameter space of layer 4 circuitry which would allow ripplets to emerge. We are indeed planning to build a new, or adapt an existing model to simulate ripplets in silico, which we prefer to defer to a future study.

There are more major concerns raised by the reviewers below, which should be addressed appropriately before we can consider publication in eLife.Should further data analysis, figures revision, and text rationale allow you to address these criticisms, we would be happy to review a revised manuscript.In the meantime, we hope you will find our referees' comments helpful. Please do not hesitate to contact us if there is anything you would like to discuss.Reviewer #1 (Recommendations for the authors):1) The authors should consider the logical flow when presenting the synchrony of the FS-FS spike burst. It would strongly support the main idea of the manuscript if the authors rephrase or/and rearrange the corresponding text and data.

We rearranged the text/figures by switching the order of Figures 3 and 4, which we hope addresses the reviewer’s concern about logical flow.

2) Please provide detailed information on the viral vector of JAWS used for the experiment of suppressing L4 RS cell firing in the methods section, and also present expression image and spike data in figures to support the idea for the subsequent experiment performed in L5.

As Figure 6 has been removed, there is no longer a need to describe the JAWS experiments.

Reviewer #2 (Recommendations for the authors):1) While the authors are generally cautious in their interpretation of the optogenetic approach and how realistic it might be, we think that this issue deserves more direct discussion. In some ways, the high degree of nearby FS synchrony induced by TC optical activation is not at all surprising given what is already known about the potency of TC inputs onto FS cells (the senior author's own work and that of many other labs), and the highly (and artificial) synchronous stimulation of many TC axons and terminals onto FS cells. The rare literature examples of high-frequency sensory evoked responses that the authors cite are arguably not simple sensory activity at all (electrical stimulation in the periphery or thalamus, both of which might lead to abnormal sensory recruitment.

We respectfully disagree with the reviewer’s suggestion that ripplets in-vivo result from “abnormal sensory recruitment.” Ripplet-like oscillations have been reported in response to strong, brief sensory stimulation, e.g., a transient whisker deflection or an auditory click in awake or lightly anesthetized rats (Jones and Barth 1999), or a tap to the finger or palm surface in awake monkeys (Baker et al. 2003), all of which are naturally occurring stimuli and are also widely used stimuli in sensory neuroscience. We made some additional textual changes in the revision to emphasize this and also to explain why ripplet-like oscillations have not been more widely reported in previous studies.

2) The results regarding the propagation of high-frequency events to layer 5 are arguably outside the main scope of the paper, and could potentially be dropped. As is, many controls that have not been provided would need to be added. For example, demonstrating a complete lack of optogenetic effects in layer 4 with Jaws to rule out that there might actually be low levels of expression that could contribute to the results. In addition, ruling out the interaction between the blue and red light stimuli in the two pathways is required, as the spectral overlap for excitation spectrum is not as clean as one would need to unambiguously interpret these results.

We agree with the sentiment of all three reviewers that the experiments shown in Figure 6 on FS bursts in layer 5 were both outside the main line of experiments and also were not conclusive, so this figure and the related text have been removed. In response to this reviewer’s specific comments, we note that the results of the optogenetic suppression experiments with JAWS were not relied upon for any of our conclusions.

Reviewer #3 (Recommendations for the authors):1) In order to prove the critical role of FS-FS interactions in timing the oscillation the authors may consider GABA-A receptor blockers and/or optogenetic inhibition of PV cells.

We find the premise of this comment puzzling, since our study argues *against* a critical role for FS-FS interactions in timing the oscillation (see, for example, the Discussion, section titled “The role of electrical or chemical FS-FS connections”). We do however argue that FS-RS connections are essential. By either model, blocking GABAergic inhibition would disrupt the oscillation, and therefore blocking GABA receptors would not help distinguish between these two competing models. Nevertheless, to demonstrate the critical need for inhibition, we added a supplementary figure (Figure 5 —figure supplement 2) showing the effect of GABAA blockers on the oscillations.

2) To gain further insights into the excitatory background of the oscillation I would propose performing RS-RS pair recordings. In this configuration, the kinetics and strength of their interactions could be measured to establish whether this is fast and strong enough to provide a drive for the oscillation. RS data similar to FS cells shown in Figure 2 (variable stimulation intensity vs jitter) could also help to elucidate the mechanisms. Using voltage-clamp experiments in RS cells could help to resolve the origin of the excitatory drive of the oscillation (TC or RS-RS interactions). Optimally experimental work would be coupled with in silico simulations to test which network activity may underly the oscillation.

The kinetics, strength and prevalence of RS-RS synaptic connections in layer 4 of barrel cortex are known from multiple previous studies, and these are now cited more broadly in the 2^nd^ paragraph of the revised Discussion. We did follow the reviewer’s recommendation and added example RS-RS recordings (in Figure 5 – figure supplement 1), to demonstrate the synchronous occurrence of E’s and I’s in neighboring RS cells.

3) In order to make predictions for future research concerning the role of this ultrafast oscillation it would be advisable to test its postsynaptic effects on targets that do not receive direct thalamic inputs (i.e. upper layer 2/3 cells). This would not only allow more insights into the generation of the oscillation but could test whether the oscillation is indeed able to select a subset of L2/3 cells as suggested. For a translaminar spread to layer 3 measuring the strength and variability of the oscillation in layer 2/3 pyramidal cells, which are not directly contacted by thalamic inputs, would greatly enhance the predictive power of the paper for future research.

Recordings from L2/3 during ripplets would indeed be very interesting and would provide important clues as to the physiological role of ripplets; however such experiments would not contribute to explaining the synaptic circuitry underlying ripplets, which is the focus of the current study, so we prefer to defer these experiments to a future study.

[Editors' note: further revisions were suggested prior to acceptance, as described below.]

We are pleased to see that the authors have addressed most of the comments made by the reviewers. The manuscript has been improved, and the unnecessary distracting parts have been removed, resulting in a smoother flow of data. However, we would like to draw your attention to a few critical comments that still need to be addressed.Firstly, the authors should include a discussion on how the strong optogenetic approaches used in this paper might or might not be relevant to the actual sensory processing in the brain circuit, given that the current data is obtained from brain slices and artificially synchronous wide-field optogenetic activation. We believe that this is an essential point to address, and we encourage you to include an appropriate discussion in the manuscript.Secondly, the authors should discuss in more detail the potential functional consequences and function of ultrafast brain oscillation in vivo, given that this manuscript focuses on the cellular nature of high-frequency oscillation in vitro. This is an excellent suggestion, and we believe that including a discussion on this topic will enhance the impact of your manuscript.We appreciate the valuable contribution that your manuscript has made, and we believe that addressing these comments on functional significance will further improve the quality and impact of your work. We have appended the full comments from the reviewers below for your reference.

In response to the latest round of review, we have made substantial textual revisions throughout the manuscript. The most substantial revisions are in response to Reviewer #2’s concern that we were confounding our optogenetic stimulation with naturalistic, physiological sensory stimuli. To alleviate this concern, we now specify, in the section of the Discussion titled “Previous observations of ripplet-like oscillations in the neocortex”, exactly what type of stimuli were used by these previous studies, making a clear distinction between peripheral nerve stimulation (used in most human and in some animal studies) and punctate tactile and auditory stimuli (used in the Barth studies, whose observations most closely resemble our own). We make a similar distinction in the relevant places in the Abstract and the Introduction.

As we state throughout the manuscript, ripplets are generated by strong synchronous activation of thalamocortical afferents. How and under what conditions is such synchrony achieved? Clearly, our optogenetic stimulation elicited the required synchrony. The reviewer seems to suggest that such synchrony will rarely, if ever, be achieved in-vivo by naturalistic stimuli, even though ripplets were elicited in the Barth studies by whisker deflections and by auditory clicks, physiological sensory stimuli which, we submit, are regularly encountered in the daily lives of rodents. We therefore take a more prudent stance and suggest in the revised document (line 430) that “Precisely what degree of thalamocortical synchrony is required for ripplet generation, and what type and intensity of sensory stimuli elicit such synchrony in vivo, is yet to be determined.”

I believe that these revisions adequately address the first point made in the editorial decision letter. The editors then add a second point, suggesting that we “discuss in more detail the potential functional consequences and function of ultrafast brain oscillation in vivo”. The potential function of ripplets was already discussed in detail in the previous submission, in the last section of the Discussion, under “Ideas and speculations: the functional significance of ripplets”. We now made additional textual revisions to this section, which I believe better clarify these ideas and suggest how spike sequences during ripplets can encode information about the sensory event that triggered them. I hope this adequately addresses this request.